# Broad protection against clade 1 sarbecoviruses after a single immunization with cocktail spike-protein-nanoparticle vaccine

Peter J. Halfmann [1,10], Kathryn Loeffler [2,10], Augustine Duffy [2,10], Makoto Kuroda[1], Jie E. Yang [3,4,5], Elizabeth R. Wright [3,4,5], Yoshihiro Kawaoka [1,6,7,8] ✉ & Ravi S. Kane [2,9] ✉

The 2002 SARS outbreak, the 2019 emergence of COVID-19, and the continuing evolution of immune-evading SARS-CoV-2 variants together highlight the need for a broadly protective vaccine against ACE2-utilizing sarbecoviruses. While updated variant-matched formulations are a step in the right direction, protection needs to extend beyond SARS-CoV-2 and its variants to include SARS-like viruses. Here, we introduce bivalent and trivalent vaccine formulations using our spike protein nanoparticle platform that completely protect female hamsters against BA.5 and XBB.1 challenges with no detectable virus in the lungs. The trivalent cocktails elicit highly neutralizing responses against all tested Omicron variants and the bat sarbecoviruses SHC014 and WIV1. Finally, our 614D/SHC014/XBB trivalent spike formulation completely protects human ACE2-transgenic female hamsters against challenges with WIV1 and SHC014 with no detectable virus in the lungs. Collectively, these results illustrate that our trivalent protein-nanoparticle cocktail can provide broad protection against SARS-CoV-2-like and SARS-CoV-1-like sarbecoviruses.

Although it has been more than three years since the identification of SARS-CoV-2, COVID-19 continues to cause considerable morbidity and mortality worldwide. Many vaccines against SARS-CoV-2 have been approved and been successful in reducing the cases of serious disease[1]. The majority of these vaccines elicit a neutralizing antibody response that targets the SARS-CoV-2 spike (S) protein[2–6]. The S protein binds to the angiotensin-converting enzyme 2 (ACE2) receptor on host cells and facilitates the fusion of the cell membrane with the viral envelope[7]. Due to its key role in viral entry, the S protein is an effective antigen for eliciting a protective immune response. Neutralizing antibodies, particularly those targeting the receptor binding domain (RBD), have been an important correlate of protection[8]. However, new SARS-CoV-2

[1]Department of Pathobiological Sciences, Influenza Research Institute, School of Veterinary Medicine, University of Wisconsin, Madison, WI 53711, USA. [2]School of Chemical & Biomolecular Engineering, Georgia Institute of Technology, Atlanta, GA 30332, USA. [3]Department of Biochemistry, University of Wisconsin, Madison, WI 53706, USA. [4]Department of Biochemistry, Cryo-EM Research Center, University of Wisconsin, Madison, WI 53706, USA. [5]Department of Biochemistry, Midwest Center for Cryo-Electron Tomography, University of Wisconsin, Madison, WI 53706, USA. [6]Division of Virology, Department of Microbiology and Immunology, Institute of Medical Science, University of Tokyo, Tokyo 108-8639, Japan. [7]The Research Center for Global Viral Diseases, National Center for Global Health and Medicine Research Institute, Tokyo 162-8655, Japan. [8]Pandemic Preparedness, Infection and Advanced Research Center (UTOPIA), University of Tokyo, Tokyo 162-8655, Japan. [9]Wallace H. Coulter Department of Biomedical Engineering, Georgia Institute of Technology, Atlanta, GA 30332, USA. [10]These authors contributed equally: Peter J. Halfmann, Kathryn Loeffler, Augustine Duffy. ✉e-mail: yoshihiro.kawaoka@wisc.edu; ravi.kane@chbe.gatech.edu

variants containing mutations that enable escape from neutralizing antibodies elicited by current vaccines or previous infection continue to emerge, such as the Omicron variants BA.5, BQ.1, and XBB[9–11]. While efforts are currently focused on developing a pan-SARS-CoV-2 vaccine to protect against current and future variants, the discovery of a large reservoir of ACE2 binding sarbecoviruses circulating in bats has prompted interest in developing pan-sarbecovirus as well as eventually pan-betacoronavirus vaccines[12].

There are two main approaches used to design vaccines to induce broad protection against highly variable viruses. One of these approaches is to focus the immune response on conserved portions of the antigen to elicit cross-reactive antibodies. For SARS-CoV-2, the highly conserved S2 subunit of the S protein has been utilized as an antigen in broadly protective vaccines[13,14]. The second approach is to incorporate proteins from different strains or subtypes of the virus into a single vaccine, thereby inducing an antibody response to each antigen and broadening the overall immune response. This approach to vaccine design is well-established and has been used in several approved vaccines targeting other pathogens such as the quadrivalent flu vaccines and the human papillomavirus 9-valent vaccine. More recently, bivalent mRNA vaccines encoding S proteins from ancestral SARS-CoV-2 and from the Omicron BA.4/BA.5 variants have been authorized for use[15].

Since authorization of the bivalent mRNA boosters, many studies have analyzed the efficacy of an additional dose of the bivalent vaccines compared to a booster of the original mRNA vaccine containing only the SARS-CoV-2 S protein in human participants. Two studies comparing the neutralization activity of sera from patients receiving either four doses of monovalent vaccines or three doses of monovalent and one dose of bivalent vaccine found that neutralization of BA.4/5, BA.4.6, and BA.2.75.2 was not significantly different between the two groups[16,17]. In contrast, other recent research has shown a slight benefit for the bivalent booster compared to the monovalent booster[18,19]. Importantly, neutralization titers against the more recent variants were noticeably lower than titers against BA.5, even for groups receiving the bivalent booster. Relative to BA.5, neutralization titers decreased 3-fold against BA.2.75.2, 4–5-fold against BQ.1.1, 6-fold against XBB, and 8.5-fold against XBB.1[18,19]. The reduction in neutralization efficiency against these new Omicron variants compared to ancestral SARS-CoV-2 is more drastic, even for groups receiving three doses of monovalent vaccine plus bivalent booster. Neutralization titers decreased 37-fold against BA.2.75.2, 41–50-fold against BQ.1.1, and 85-100-fold against XBB.1[9,19]. Together these results present a clear need for an updated vaccine against recent and emerging Omicron variants.

While vaccines that provide pan-SARS-CoV-2 immunity would be of immediate interest in the context of the continuing pandemic, there is particular interest in the development of pan-sarbecovirus vaccines that elicit a protective antibody response towards non-SARS-CoV-2 sarbecoviruses. Tan et al.[20] found that the sera of individuals who had been previously infected with SARS-CoV-1 and then vaccinated against SARS-CoV-2 displayed enhanced neutralization against a panel of sarbecoviruses including both SARS-CoV-2 and SARS-CoV-1, suggesting that a multivalent vaccine could be a promising approach. Indeed, several groups are developing multivalent vaccines that combine protein antigens from different sarbecoviruses. Cohen et al. evaluated the performance of a mosaic protein nanoparticle vaccine displaying the receptor-binding domains (RBDs) from eight sarbecoviruses in mice and hamsters[21,22]. Two doses of the mosaic RBD nanoparticle vaccine elicited high levels of broadly neutralizing antibodies and protected against challenges with SARS-CoV-2 beta and delta variants and SARS-CoV-1. Neutralization titers against the most recent Omicron variants such as BA.2.75, XBB, and BQ.1 were not tested because these variants had not yet been reported at the time of the study. While these results are promising, the high variability of the RBD and immune

evasion displayed by variants may present a challenge in eliciting pan-sarbecovirus immunity. Du et al. used the full S protein trimer in a bivalent protein subunit vaccine containing S-614G and BA.1 S[23]. Three doses of the bivalent vaccine elicited higher neutralization titers against both early SARS-CoV-2 variants and Omicron variants BA.1 and BA.2 compared with vaccines containing only S-614G or BA.1 S. In a similar strategy, Binkkemper et al. displayed the S protein of ancestral SARS-CoV-2 and SARS-CoV-1 on a virus-like particle (VLP) in both a mosaic formulation with both antigens on the same particle as well as a cocktail formulation consisting of mixtures of particles each presenting only one type of antigen[24]. Three doses of the mosaic and cocktail vaccines induced higher neutralization titers against SARS-CoV-1 and WIV1 than three doses of the VLP vaccine containing only SARS-CoV-2 S, although little improvement was seen in the neutralization titers against SHC014, BA.1, and BA.4/5. The performance of the vaccines against a viral challenge was not evaluated. Collectively, these results suggest that a cocktail vaccine comprising carefully selected mixtures of S protein antigens represents a promising strategy for developing a pan-sarbecovirus vaccine. However, there is currently little data demonstrating the in vivo efficacy of S-protein-based vaccines against BA.5 and the more recent Omicron variants.

Protein nanoparticles have emerged as attractive platforms for the display of S protein antigens. Brouwer et al. generated two component protein nanoparticles displaying stabilized prefusion SARS-CoV-2 S proteins that protected vaccinated macaques from a challenge with SARS-CoV-2[25]. Joyce et al. showed that adjuvanted SARS-CoV-2 S protein-ferritin nanoparticle vaccines protected non-human primates from a challenge with SARS-CoV-2[26]. Weidenbacher et al. reported that adjuvanted ferritin nanoparticle vaccines displaying a truncated form of the SARS-CoV-2 S protein ectodomain elicited a broad neutralizing antibody response in non-human primates[27]. Hutchinson et al. designed self-assembling protein nanoparticles displaying multiple S protein antigens that protected mice from a challenge with MERS-CoV[28]. As described above, Brinkkemper et al. designed nanoparticles presenting mixtures of the SARS-CoV-1 and SARS-CoV-2 S proteins[24].

Previously, we developed a nanoparticle vaccine displaying multiple copies of the SARS-CoV-2 614D S protein (VLP-S)[29]. A single immunization with VLP-S in hamsters elicited high neutralizing antibody titers and protected Syrian hamsters from a challenge with an early isolate of SARS-CoV-2 with no infectious virus detected in the lungs. We reasoned that this VLP-S platform would enable the design of a cocktail vaccine that elicits a broad neutralizing antibody response against human-ACE2-binding Clade 1 sarbecoviruses with pandemic potential (Fig. 1a)—not just SARS-CoV-2 variants but also SARS-CoV-1 and related bat sarbecoviruses. We first generated a panel of VLPs, each displaying a single S protein from various sarbecoviruses (614D BA.1, BA.5, BA.2.75.2, XBB, SARS-CoV-1, and SHC014) and assessed their immunogenicity in hamsters. Based on an analysis of the antigenic landscape, we selected bivalent and trivalent mixtures and further characterized the breadth of the neutralizing antibody response elicited by immunization as well as the ability to protect from challenges with Omicron variants XBB.1 and BA.5 as well as bat coronaviruses (CoVs) SHC014 and WIV1. We demonstrate that the selected trivalent formulations consistently elicited robust neutralization titers against several Omicron variants, SHC014, and WIV1. Additionally, immunization with this cocktail vaccine provided complete protection against challenges with BA.5, XBB.1, SHC014, and WIV1, with no detectable viral titers in the lungs. Collectively, these results strongly suggest that VLP-S cocktail vaccines have the potential to provide broad protection against all significant Clade 1 sarbecoviruses.

## Results
### Selection of S proteins for immunization
First, an appropriate mix of S proteins had to be selected for a preliminary evaluation of immunogenicity. Sarbecoviruses can be divided

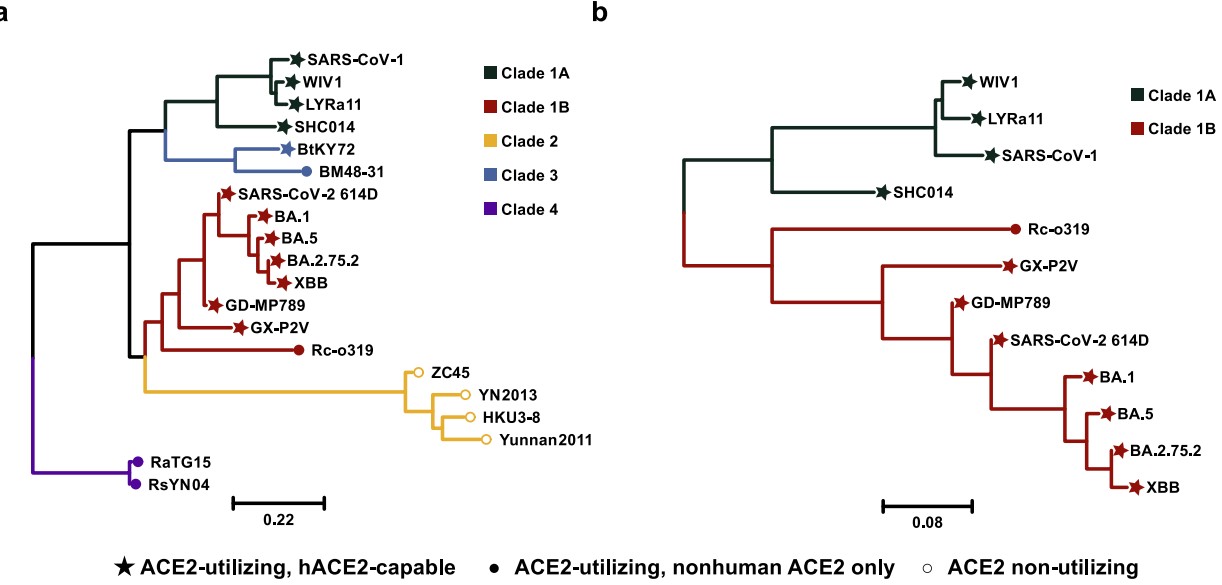

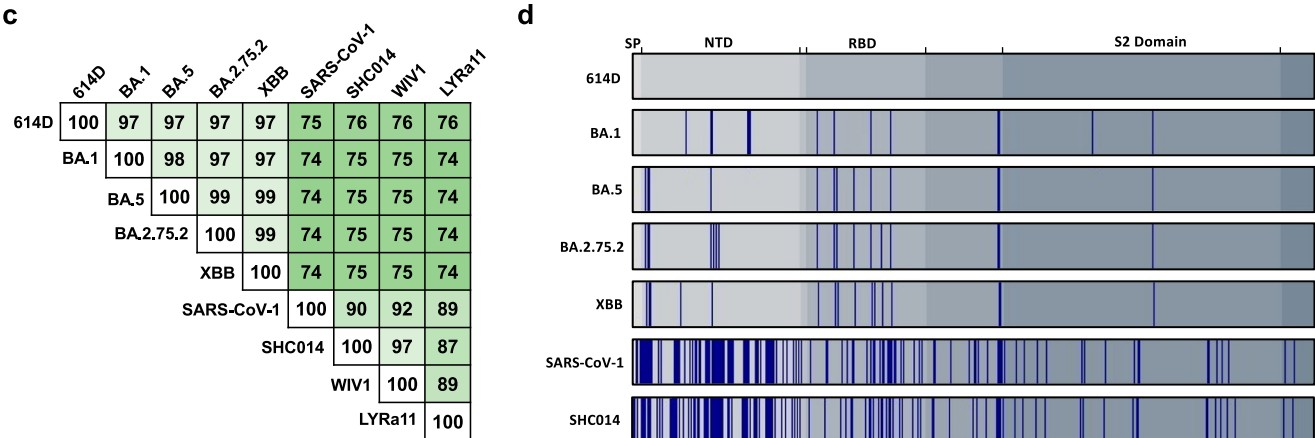

**Fig. 1 | Analysis of Sequences of *Sarbecovirus* RBD and S Proteins. a** Amino acid phylogenetic tree of *Sarbecovirus* RBDs. Scale bar represents the number of mutation events per residue. Node shapes denote receptor usage (★: ACE2-utilizing, hACE2-capable; ●: ACE2-utilizing, nonhuman ACE2 only; ○: ACE2 non-utilizing), with hACE2-capable referring to viruses able to infect cells expressing human ACE2 in vitro. Clade assignments are indicated by branch and node color (Clade 1A: dark green; Clade 1B: dark red; Clade 2: yellow; Clade 3: light blue; Clade 4: purple). Accession numbers are provided in Supplementary Table 2.

**b** Amino acid phylogenetic tree of *Sarbecovirus* RBDs in Clades 1A and 1B. Node shapes and branch colors are the same as in Fig. 1a. **c** Sequence identity of S proteins selected for immunization. White indicates 100% sequence identity, and darker shades of green indicate lower percentages. **d** Location of amino acid changes of selected S proteins compared to SARS-CoV-2 614D. Gray areas indicate different domains of the S protein, and blue bars indicate amino acid changes in relation to 614D. SP signal peptide, NTD N-terminal domain, RBD receptor binding domain.

into multiple clades based on their RBDs (Fig. 1a)[30]. Clade 1 contains the sarbecoviruses known to cause disease in humans, including SARS-CoV-1 (Clade 1A) and SARS-CoV-2 (Clade 1B), as well as the sarbecoviruses considered to have the greatest potential to cause future pandemics[12,31–33]. Our priority in this work, therefore, was to generate a cocktail vaccine that elicited a broad neutralizing antibody response against viruses from Clades 1A and 1B. We selected S antigens based on an analysis of the phylogenetic tree (Fig. 1a, b) and their amino acid sequence homology (Fig. 1c, d). As potential S antigens from Clade 1A, we chose SARS-CoV-1 and bat CoV SHC014 due to their low amino acid homology with each other relative to other Clade 1A S proteins (Fig. 1c) and the reduced neutralization of SHC014 by SARS-CoV-1 convalescent sera compared to WIV1, another Clade 1A bat coronavirus[20]. As candidate antigens from Clade 1B, we selected the ancestral SARS-CoV-2 S protein (614D) as well as the S proteins from four different Omicron variants: BA.1, BA.5, BA.2.75.2, and XBB.

## Generation and characterization of S nanoparticle-based vaccines

We previously developed VLPs displaying the 614D HexaPro[34] S protein[29]. The VLPs are composed of 90 homodimers of the bacteriophage MS2 coat protein[35] with an AviTag inserted into a surface loop that self-assemble into an icosahedral structure. AviTagged MS2 VLPs were biotinylated and then mixed with a large excess of streptavidin (SA) to produce streptavidin-coated VLPs (MS2-SA). Biotinylated S proteins of each sarbecovirus were also produced as previously described. In brief, HexaPro variants of each S protein with a C-terminal trimerization domain, AviTag, and his-tag were expressed in Expi293F mammalian cells. S proteins were purified by immobilized metal affinity chromatography (IMAC), biotinylated in vitro, and purified by size exclusion chromatography (SEC). VLP-S particles were produced by mixing the appropriate ratio of MS2-SA to biotinylated S protein. This ratio corresponded to the mixture of MS2-SA and S with

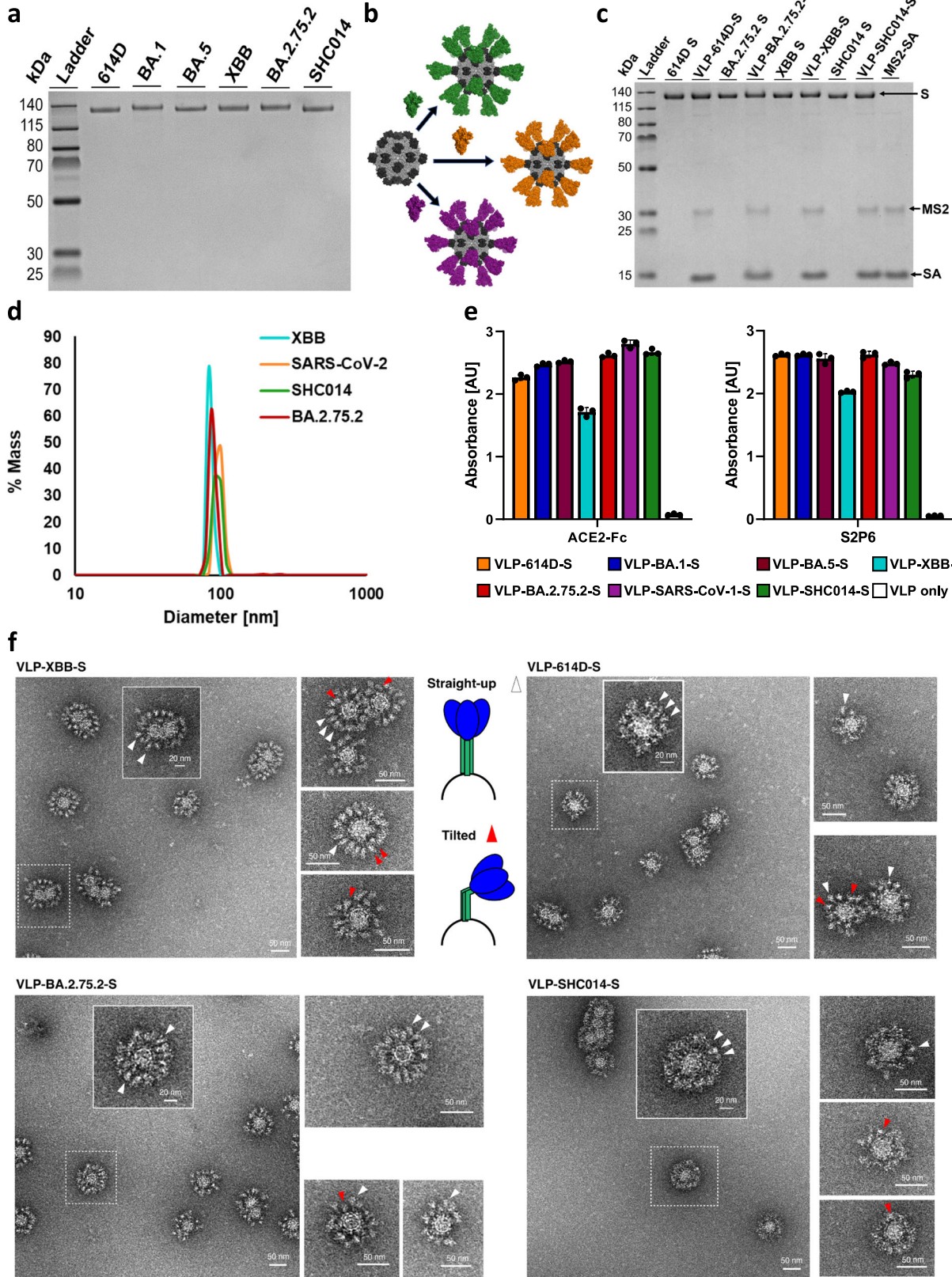

the lowest amount of MS2-SA that did not produce a peak indicating excess S on an SEC chromatogram.

The S antigens and VLP-S vaccines were characterized by several techniques. Prior to displaying the S proteins on MS2-SA, the purity of the biotinylated S was verified by SDS polyacrylamide gel electrophoresis (SDS-PAGE) (Fig. 2a). Biotinylated S proteins were mixed

individually with MS2-SA to produce VLP-S (Fig. 2b) and then characterized by SDS-PAGE (Fig. 2c). The size of the VLP-S particles was measured by dynamic light scattering (DLS). The diameter was measured to be 90–100 nm, consistent with prior characterization, and was consistent for all VLP-S nanoparticles (Fig. 2d and Supplementary Fig. 1).

**Fig. 2 | Characterization of S and VLP-S. a** SDS-PAGE characterization of biotinylated S proteins. The unprocessed gel is shown in Supplementary Fig. 3. This gel was run twice from the same preparation for each sample with similar results. **b** Schematic of the attachment of various biotinylated S proteins to MS2-SA. (MS2: light gray, PDB 2MS2; SA: dark gray, PDB 3RY2; S: green/orange/purple, PDB 6VSB) (**c**) SDS-PAGE gel of S and VLP-S for 614D, BA.2.75.2, XBB, and SHC014. Each VLP-S has been boiled to disrupt the streptavidin-biotin conjugation. The unprocessed gel is shown in Supplementary Fig. 3. This gel was run twice from the same preparation for each sample with similar results. **d** Characterization of VLP-614D-S (orange), VLP-SHC014-S (green), VLP-BA.2.75.2-S (red), and VLP-XBB-S (cyan) by dynamic light scattering. **e** Characterization of the binding of ACE2-Fc and S2P6 antibody to all VLP-S. (mean ± SD, *n* = 3: one independent assay with three technical replicates). Bar color identifies each VLP-S sample (VLP-614D-S: orange; VLP-BA.1-S: dark blue; VLP-BA.5-S: brown; VLP-XBB-S: cyan; VLP-BA.2.75.2: red; VLP-SARS-CoV-1-S: purple; VLP-SHC014-S: green; VLP only: white). **f** Characterization of VLP-XBB-S, VLP-614D-S, VLP-BA.2.75.2-S, and VLP-SHC014-S by negative stain transmission electron microscopy. Arrowheads ▲ indicate S proteins on the VLP surface, with white arrowheads indicating straight-up spike proteins and red arrowheads indicating tilted spike proteins. At least 70 images were collected and analyzed from one VLP-S preparation for each sample with similar results.

Next, we measured the binding of ACE2-Fc (a part of the ACE2 receptor ectodomain linked to an Fc domain), two RBD-binding antibodies CR3022 and S309, and an S2-binding antibody S2P6 to the S proteins and VLP-S particles through enzyme-linked immunosorbent assay (ELISA). Binding remained high for all S proteins when attached to the MS2-SA VLP, indicating that the antigens retained their proper structure and folding (Fig. 2e and Supplementary Fig. 2).

Finally, we performed negative-stain transmission electron microscopy (NS-TEM) on individual VLP-S (Fig. 2f). Consistently, all four VLP-S variants displayed a size range of 70–80 nm in diameter and had a high coating efficiency of glycoproteins (above 95%). A closer examination of VLP-S reveals that S-glycoproteins did not all protrude perpendicularly from the VLP surface (white arrowheads, straight-up morphology). Some were tilted, noted with red arrowheads (Fig. 2f). Often, both straight-up and tilted glycoproteins were present on one decorated MS2-SA. While it is possible that NS-TEM could introduce conformational changes in the staining process, the observed tilted conformation of S-glycoproteins is consistent with previous cryo-electron tomography (cryo-ET) studies where SARS-CoV-2 S trimers can be highly tilted towards the membrane on the native viral particles[36]. VLP-614D-S, VLP-XBB-S, and VLP-BA.2.75.2-S had similar morphologies with some distinct individual morphological features, possibly due to a variation in glycoprotein number and packing on the surface of the VLPs. In comparison, we observed a denser and fuzzier morphology on the VLPs presenting the bat sarbecovirus S, VLP-SHC014-S.

## Monovalent VLP-S vaccines elicit high neutralization titers against similar sarbecoviruses

We first assessed the immunogenicity and protective efficacy of each of the VLP-S vaccines separately. Syrian hamsters were immunized with VLP-control (MS2-SA only) or VLPs displaying either the 614D, BA.1, BA.5, BA.2.75.2, XBB, SARS-CoV-1, or SHC014 S protein, all adjuvanted with Alhydrogel.

Neutralization was characterized using a focus reduction neutralization test with titers reported as the reciprocal of the dilution at which the number of foci was reduced by 50% (FRNT50). FRNT50 values were determined four weeks after immunization against an early isolate of SARS-CoV-2 (S-614G), the Omicron variants BA.1, BA.5, and XBB.1, and bat CoVs SHC014 and WIV1 (Fig. 3a). Immunization with Clade 1A VLP-S (SARS-CoV-1 and SHC014) did not elicit detectable neutralization titers against Clade 1B viruses. Groups immunized with Clade 1B VLP-S (614D, BA.1, BA.2.75.2, and XBB) displayed only minor neutralizing activity against SHC014 and no neutralizing activity against WIV1. VLP-614D-S elicited high neutralization titers against 614 G (geometric mean value of 5080) though titers decreased against the Omicron variants with no detectable neutralization titers against XBB.1. Both SHC014 and SARS-CoV-1 immunogens elicited appreciable titers against SHC014 and WIV1. VLP-SHC014-S elicited higher neutralizing antibody titers against WIV1 compared to VLP-SARS-CoV-1-S, consistent with the WIV1 S being more homologous to the SHC014 S than to the SARS-CoV-1 S (Fig. 1c).

## VLPs displaying the S Protein from SARS-CoV-2 variants protect against BA.5

Six weeks after immunization, hamsters were intranasally inoculated with $10^5$ plaque-forming units (pfu) of BA.5 (hCoV-19/Japan/TY41-702/2022). The hamsters were sacrificed three days after inoculation, and the viral titers in the lungs were measured by plaque assay. The reduction of virus titers in the lungs after BA.5 challenge in immunized hamsters (Fig. 3b) was consistent with the observed trends in neutralization activity (Fig. 3a). Immunization with VLP-614D-S or any of the four Omicron VLP-S vaccines, all of which elicited high neutralizing antibody titers against BA.5, provided full protection from a BA.5 challenge as indicated by undetectable levels of virus in the lungs. However, immunization with either VLP-SHC014-S or VLP-SARS-CoV-1-S—neither of which elicited any detectable neutralizing antibodies against BA.5—did not protect against a BA.5 challenge, with viral lung titers comparable to those in hamsters immunized with the control VLP (Fig. 3b). Similar trends were observed in viral titers in the nasal turbinates as VLP-614D-S and all of the Omicron VLP-S vaccines significantly reduced virus levels, while hamsters vaccinated with VLP-SHC014-S and VLP-SARS-CoV-1-S had viral levels comparable to those of the control hamsters (Fig. 3c).

## Antigenic cartography to select vaccine cocktails

Next, we applied antigenic cartography to our neutralization data (Fig. 3d). Using neutralization titers, this technique displays the assay viruses and the antisera from vaccinated hamsters on a two-dimensional map where viruses with similar antigenicity are located near each other, and antisera are located near the viruses that they best neutralize. Two virus populations can be seen in our antigenic map, with the Clade 1A viruses WIV1 and SHC014 clustering together, and the Clade 1B SARS-CoV-2 variants forming a separate elongated cluster. Antisera also appear to form similar groupings; sera from VLP-SHC014-S and VLP-SARS-CoV-1-S form a Clade 1A cluster; VLP-BA.5-S, VLP-BA.2.75.2-S, and VLP-XBB-S sera are all similarly co-located; and VLP-614D-S and VLP-BA.1-S sera stand by themselves, albeit nearby the other SARS-CoV-2 VLP-S antisera.

We next used the antigenic map to help select VLP-S conjugates for inclusion in the candidate cocktail vaccines based on their corresponding antisera. We aimed to incorporate the fewest VLP-S conjugates whose antisera would collectively be able to neutralize all of the viruses on the map. Compared to VLP-SARS-CoV-1-S, VLP-SHC014-S elicited higher neutralization titers against both bat CoVs tested (Fig. 3a); as such, VLP-SHC014-S was chosen as the Clade 1A representative. VLP-614D-S was chosen to cover the early SARS-CoV-2 variants as it outperforms VLP-BA.1-S against 614 G (Fig. 3a) and is further from the two remaining clusters (Fig. 3c). Based on Tan et al.'s findings that sera from SARS-CoV-1 patients who were vaccinated against SARS-CoV-2 could neutralize both Clade 1A and Clade 1B viruses[20], we expected that a bivalent mixture of VLP-614D-S and VLP-SHC014-S should cover the majority of Clade 1 sarbecoviruses. However, due to decreasing neutralization against BA.5 and XBB.1 and the continuing emergence of increasingly distant Omicron

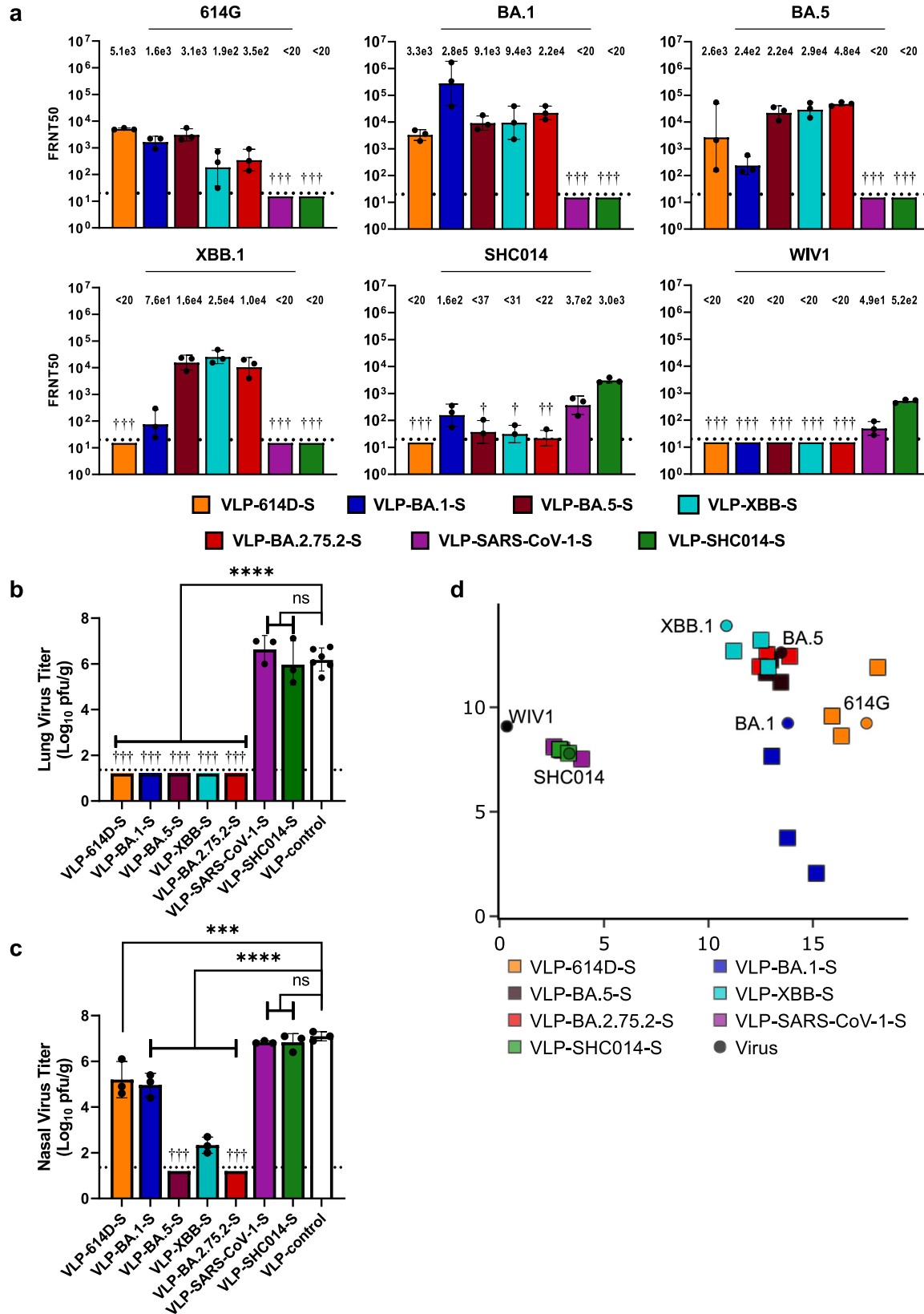

variants with some even arguing that the Omicron lineage comprises a separate serotype from pre-Omicron variants[37,38], we anticipated that the inclusion of an additional Omicron variant VLP-S would be required. Accordingly, we also generated trivalent vaccines containing either VLP-BA.2.75.2-S or VLP-XBB-S in addition to VLP-614D-S and VLP-SHC014-S.

**Trivalent mixtures elicit a broadly neutralizing antibody response and protect hamsters against challenges with BA.5, XBB.1, WIV1, and SHC014**

Hamsters were immunized with the bivalent formulation containing VLP-614D-S and VLP-SHC014-S only or with a trivalent vaccine containing VLP-614D-S, VLP-SHC014-S and either VLP-BA.2.75.2-S or VLP-

**Fig. 3 | Protective efficacy of individual VLP-S.** For Fig. 3a–c, ● or † indicate data from individual hamsters and bar color identifies each VLP-S vaccine (VLP-614D-S: orange; VLP-BA.1-S: dark blue; VLP-BA.5-S: brown; VLP-XBB-S: cyan; VLP-BA.2.75.2: red; VLP-SARS-CoV-1-S: purple; VLP-SHC014-S: green). **a** FRNT50 neutralization titers against 614 G, BA.5, BA.1, XBB.1, SHC014, and WIV1 after a single immunization with the individual VLP-S (geometric mean with geometric SD, $n = 3$: sera from 3 hamsters). † - No neutralization titers detected. Detection limit (dotted line) = 20. **b** Viral titers in the lungs of hamsters immunized with VLP-S or VLP-control three days after infection with BA.5 (mean with SD, $n = 3$: tissue from 3 hamsters). ****$P < 0.0001$, ns - not significant (VLP-control vs VLP-SARS-CoV-1-S: $P = 0.6265$; VLP-control vs VLP-SHC014-S: $P = 0.9953$) determined by a one-way analysis of variance (ANOVA) and Dunnett's post-hoc multiple comparisons test † - No infectious virus was detected in the lungs of immunized hamsters. Detection limit (dotted line) = 1.3 $\log_{10}$ pfu/g. **c** Viral titers in the nasal turbinates of hamsters immunized with VLP-S or VLP-control three days after infection with BA.5 (mean with SD, $n = 3$: tissue from 3 hamsters). ****$P < 0.0001$, ***$P = 0.0001$, ns – not significant (VLP-control vs VLP-SARS-CoV-1-S: $P = 0.9216$; VLP-control vs VLP-SHC014-S: $P = 0.9976$), determined by a one-way ANOVA with Dunnett's multiple comparisons test. † - No infectious virus was detected in the nasal turbinates of immunized hamsters. Detection limit (dotted line) = 1.3 log10 pfu/g. **d** Antigenic map generated from neutralization titers of hamsters vaccinated with monovalent VLP-S. Antisera are represented by □, and viruses are represented by ●. One unit corresponds to a two-fold reduction in neutralization titer. Sera from 3 hamsters for each vaccine condition, with each □ representing sera from one hamster. □ color identifies each VLP-S vaccine (VLP-614D-S: orange; VLP-BA.1-S: dark blue; VLP-BA.5-S: brown; VLP-XBB-S: cyan; VLP-BA.2.75.2: red; VLP-SARS-CoV-1-S: purple; VLP-SHC014-S: green). ● color and a text label identifies viruses (614 G: orange; BA.1: dark blue; BA.5: brown; XBB.1: cyan; SHC014: green; WIV1: black).

XBB-S. Each vaccine contained 15 μg of each S displayed on MS2-SA VLPs adjuvanted with Alhydrogel. To serve as a point of comparison, an additional group of hamsters was immunized with a single dose (30 μg) of the Pfizer-BioNTech bivalent vaccine.

We first characterized the breadth of the neutralizing antibody response against the same panel of sarbecoviruses as before. All three formulations elicited comparably high titers of neutralizing antibodies against 614 G and BA.5 (Fig. 4a). The two trivalent vaccines incorporating a newer Omicron strain elicited high neutralizing titers against XBB.1 as well; however, the bivalent vaccine showed reduced neutralization of XBB.1. This further supports our observation that the inclusion of a third Omicron antigen would be necessary to elicit a broad neutralizing response against the newest variants. Next, we assessed the neutralization titers against the two SARS-like bat CoVs, SHC014 and WIV1. As anticipated, all three cocktail vaccines elicited similar levels of neutralizing antibodies against the two bat viruses. To compare, hamsters immunized with 30 μg of the Pfizer-BioNTech bivalent vaccine did produce neutralizing antibodies against an early SARS-CoV-2 isolate (S-614G) and BA.5 as expected given the composition of the vaccine. Like our bivalent 614D/SHC014 vaccine, neutralization titers were significantly reduced against XBB.1 (3 out of 4 hamsters had no detectable titers), but unlike any of our bivalent or trivalent vaccines, no detectable neutralization titers were observed against the Clade 1A bat CoVs SHC014 and WIV1. These results further support the inclusion of antigens from both XBB-like and Clade 1A CoVs in a broadly protective cocktail vaccine.

Next, we tested the ability of the bivalent and trivalent vaccine formulations to protect hamsters from challenges with the Omicron variants BA.5 and XBB.1. Six weeks after immunization, hamsters were inoculated intranasally with $10^5$ pfu of BA.5 or XBB.1 and the viral titers in the lungs and nasal turbinates were quantified three days after inoculation. All three vaccine cocktails provided full protection against both BA.5 and XBB.1 (Fig. 4b) challenges as shown by the lack of detectable viral titers in the lungs of immunized hamsters, while hamsters immunized with the control VLP had high viral titers. The bivalent vaccine provided complete protection against the XBB.1 challenge despite eliciting significantly lower neutralizing antibody titers against XBB.1 than the trivalent vaccines. Similar trends were also seen for nasal titers. The VLP-614D-S and VLP-SHC014-S bivalent cocktail significantly reduced nasal virus titers, while hamsters immunized with either trivalent cocktail had undetectable nasal titers (Fig. 4d).

While neutralizing antibody titers against SARS-CoV-2 have been identified as a correlate of protection against symptomatic disease[8,39], even low neutralizing titers can be sufficient for protection. High neutralization titers elicited by vaccination are still desirable, since they are likely to provide greater protection against viral escape; neutralization titers against viral variants are generally lower than those against the vaccine strain. The inclusion of an Omicron antigen in the cocktail is thus justified, particularly in light of the continuing emergence of new Omicron variants. Meanwhile, a single immunization with 30 μg of the Pfizer-BioNTech bivalent vaccine significantly decreased lung and nasal titers after challenge with XBB.1 (Fig. 4c, e).

To further examine the breadth of protection conferred by our trivalent vaccines, we vaccinated transgenic hamsters expressing human ACE2 (hACE2) with our vaccine formulation comprised of VLP-614D-S, VLP-SHC014-S, and VLP-XBB-S. Hamsters were then challenged six weeks after immunization with either WIV1 or SHC014, and viral lung and nasal turbinate titers were determined three days later. In both cases, vaccinated hamsters had no detectable virus in their lungs while control hamsters had high viral titers (Fig. 5a). Nasal turbinate titers were also significantly reduced in vaccinated transgenic hamsters in both cases (Fig. 5b), though the magnitude of this reduction was weaker than for the Clade 1B viruses.

## Discussion

Taken together, these results show that our trivalent VLP-S vaccine formulation can both elicit a broadly neutralizing antibody response against a panel of Clade 1 sarbecoviruses and also provide complete protection in lungs of hamsters against challenges by both Clade 1A (SHC014 and WIV1) and Clade 1B (BA.5 and XBB.1) viruses.

While we are encouraged by these results, there are several additional avenues that would be interesting to explore in future work. Enhancing mucosal immunity might not only enhance protection against viral infection, but also decrease viral transmission[40]. Intranasal vaccination against SARS-CoV-2 has been explored with several platforms, including mRNA-lipid nanoparticles[41], nanoparticles displaying the RBD[42], live attenuated influenza viruses also encoding the RBD[43], adenovirus-vectored vaccines[44], and by using an intranasal boost with the unadjuvanted spike protein[40]. The adaptation of our platform for intranasal delivery could be a promising avenue for improving the mucosal response. Characterizing the longevity of protection would also be an interesting avenue for future research. It would be particularly interesting to determine whether stronger mucosal immunity results in more durable protection against symptomatic disease.

Secondly, while the focus of this paper was on protecting against viruses in Clades 1A and 1B, extending protection to further sarbecovirus clades would be interesting to explore. Significant differences in their receptor binding domains are the primary basis for virus classification into clades, with Clade 2 viruses being unable to use ACE2 as an entry receptor and Clade 3 & 4 viruses harboring one deletion relative to Clade 1[45]. Nevertheless, some Clade 2 viruses (a proposed Clade 2A)[46] and several Clade 3 viruses[47] may be capable of infecting human cells, albeit in some cases only with exogenous protease treatment. RBD- and NTD-focused humoral immunity is unlikely to be cross-reactive between clades, as inter-clade RBD and NTD amino acid identity percentages are approximately 65–75% and 45–55%, respectively. As such, the inclusion of additional antigens to cover these clades may be necessary. Using the approach described in this work,

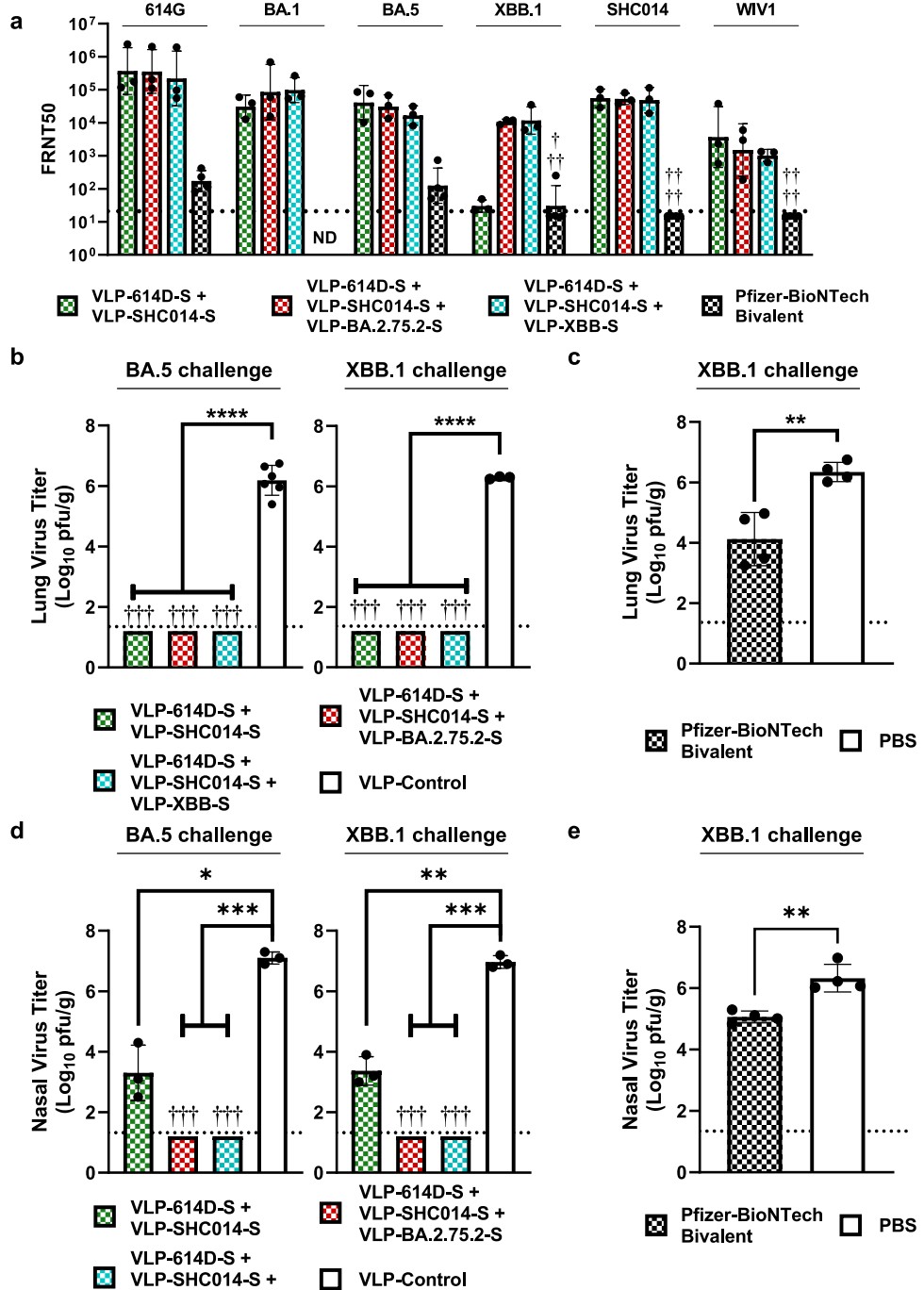

**Fig. 4 | Protective efficacy of bivalent and trivalent VLP-S mixtures in Syrian hamsters.** ● or † indicate data from individual hamsters. Bar color indicates vaccine group (VLP-614D-S/VLP-SHC014-S: green pattern; VLP-614D-S/VLP-SHC014-S/VLP-BA.2.75.2-S: red pattern; VLP-614D-S/VLP-SHC014-S/VLP-XBB-S: cyan pattern; VLP control or PBS: white). **a** FRNT50 neutralization titers against indicated viruses after immunization with VLP-S mixtures or Pfizer-BioNTech Bivalent mRNA vaccine. (geometric mean with geometric SD, $n = 3$: sera from 3 hamsters for VLP-S; $n = 4$: sera from 4 hamsters for Pfizer-BioNTech vaccine). † - No neutralization titers detected. ND – Not determined. Detection limit (dotted line) = 20. **b** Viral titers in the lungs of hamsters immunized with VLP-S cocktails or VLP-control three days after infection with BA.5 or XBB.1 (mean with SD, $n = 3$: tissue from 3 hamsters for VLP-S and XBB.1 VLP-control; $n = 6$: tissue from 6 hamsters for BA.5 VLP-control). ****$P < 0.0001$, determined by a Brown-Forsythe and Welch ANOVA test and Dunnett's T3 multiple comparisons between groups. For Fig. 4b–e, † - No infectious virus was detected; detection limit (dotted line) = 1.3 $\log_{10}$ pfu/g. **c** Viral titers in the lungs of hamsters immunized with Pfizer-BioNTech Bivalent mRNA vaccine or PBS three days after infection with XBB.1 (mean with SD, $n = 4$: tissue from 4 hamsters). **$P = 0.0031$, determined by two-tailed unpaired Student's $t$ test. **d** Viral titers in the nasal turbinates of hamsters immunized with VLP-S cocktails or VLP-control three days after infection with BA.5 or XBB.1 (mean with SD, $n = 3$: tissue from 3 hamsters for VLP-S and XBB.1 VLP-control; $n = 6$: tissue from 6 hamsters for BA.5 VLP-control). *$P = 0.0396$, **$P < 0.01$ (VLP-control vs VLP-614D-S/VLP-SHC014-S: $P = 0.0040$; VLP-control vs VLP-614D-S/VLP-SHC014-S/VLP-XBB-S: $P = 0.0010$; VLP-control vs VLP-614D-S/VLP-SHC014-S/VLP-BA.2.75.2-S: $P = 0.0010$), ***$P = 0.0008$, determined by a Brown-Forsythe and Welch ANOVA test and Dunnett's T3 multiple comparisons between groups. **e** Viral titers in the nasal turbinates of hamsters immunized with Pfizer-BioNTech Bivalent mRNA vaccine or PBS three days after infection with XBB.1 (mean with SD, $n = 4$: tissue from 4 hamsters). **$P = 0.0021$, determined by two-tailed unpaired Student's $t$ test.

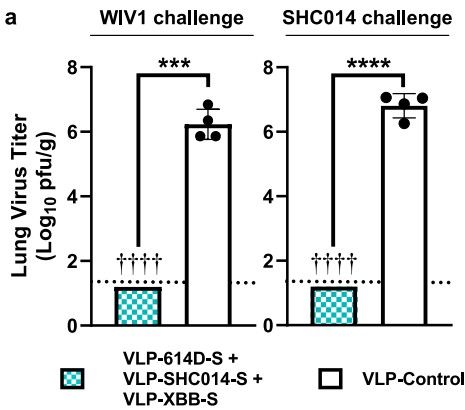
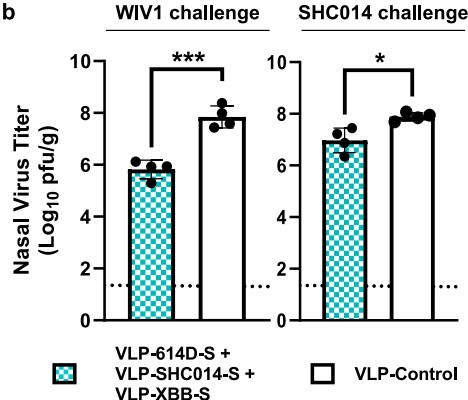
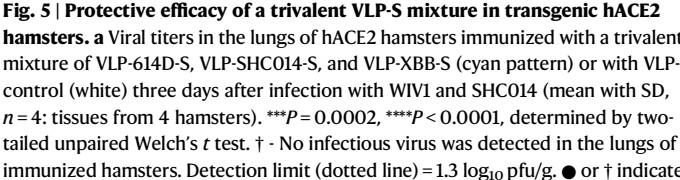

**Fig. 5 | Protective efficacy of a trivalent VLP-S mixture in transgenic hACE2 hamsters. a** Viral titers in the lungs of hACE2 hamsters immunized with a trivalent mixture of VLP-614D-S, VLP-SHC014-S, and VLP-XBB-S (cyan pattern) or with VLP-control (white) three days after infection with WIV1 and SHC014 (mean with SD, $n = 4$: tissues from 4 hamsters). ***$P = 0.0002$, ****$P < 0.0001$, determined by two-tailed unpaired Welch's $t$ test. † - No infectious virus was detected in the lungs of immunized hamsters. Detection limit (dotted line) = 1.3 log$_{10}$ pfu/g. ● or † indicate

data from individual hamsters. **b** Viral titers in the nasal turbinates of hACE2 hamsters immunized with a trivalent mixture of VLP-614D-S, VLP-SHC014-S, and VLP-XBB-S (cyan pattern) or with VLP-control (white) three days after infection with WIV1 and SHC014 (mean with SD, $n = 4$: tissues from 4 hamsters). ***$P = 0.0003$, *$P = 0.0108$, determined by a two-tailed unpaired Student's $t$ test. Detection limit (dotted line) = 1.3 log$_{10}$ pfu/g. ● indicate data from individual hamsters.

antigenic cartography should help us identify the minimal mixture of antigens required to elicit broad protection against sarbecoviruses.

Given that protein-nanoparticle vaccines have been approved for clinical use[48], commercialization of this vaccine platform could be possible in the future. Moreover, the optimal components for a cocktail vaccine that are suggested by antigenic cartography may be platform-agnostic and could therefore be applied not just to other protein nanoparticle platforms, but also to other modalities such as mRNA-based vaccines. Indeed, the selection of an XBB strain (XBB.1.5) for incorporation in the recently approved monovalent vaccine is consistent with our results and it would be interesting to explore the protective efficacy of trivalent mRNA vaccines based on the compositions identified in this work against Clade 1 sarbecoviruses.

We did not test mosaic vaccines as part of the current work. While the underlying concept is interesting, a recent study reported a head-to-head comparison between cocktail and mosaic approaches for spike protein-nanoparticle vaccines and did not identify a significant difference in neutralization titers between the two approaches[24]. A cocktail vaccine is also aligned with the hypothesis motivating this work—that if the mixture of antigens comprising the cocktail is carefully selected by characterizing the antigenic landscape, the antibody response elicited by each component would collectively result in a broadly protective polyclonal antibody response. It would be interesting to revisit the mosaic idea in future work.

**Limitations of this study**

This study used hamsters to evaluate the immunogenicity and efficacy of our vaccines because hamsters are susceptible to infection by sarbecoviruses. While hamsters are valuable models for sarbecovirus infection and prevention of disease, effectiveness in the hamster model may not translate into real-world effectiveness in humans. As such, this work is only one step towards the goal of developing a pan-sarbecovirus vaccine for use in humans. Furthermore, one important consideration not accounted for in this study is the effect of pre-existing immunity on protection outcomes. While a naïve animal might respond equally to all components of a vaccine, an animal with pre-existing immunity might instead be biased towards one component to which it has been exposed to before[49–51]. With much of the world's population having prior exposure to SARS-CoV-2 or its variants through infection and/or vaccination, this immunological imprinting may impact the design of effective broadly protective vaccines. Finally, while we evaluated our vaccines against a panel of Clade

1 sarbecoviruses for both neutralization assays and viral challenges, our panel was certainly not a comprehensive survey of all known viruses. Nevertheless, we believe that this study is a valuable step towards designing a pan-sarbecovirus vaccine and may help guide future studies that address these limitations.

## Methods

### Expression and purification of S proteins

DNA encoding HexaPro[34] prefusion-stabilized versions of the S ecto-domains of SARS-CoV-2 614D (YP_009724390.1 [https://www.ncbi.nlm.nih.gov/protein/YP_009724390.1/], residues 1–1208), Omicron BA.1 (UFO69279.1 [https://www.ncbi.nlm.nih.gov/protein/UFO69279.1/], residues 1–1205), Omicron BA.5.5 (UPI46221.1 [https://www.ncbi.nlm.nih.gov/protein/UPI46221.1/], residues 1–1203), Omicron BA.2.75.2 (UVJ48842.1 [https://www.ncbi.nlm.nih.gov/protein/UVJ48842.1/], residues 1–1205), Omicron XBB (UZS22117.1 [https://www.ncbi.nlm.nih.gov/protein/UZS22117.1/], residues 1–1204), SARS-CoV-1 Urbani (P59594 [https://www.ncbi.nlm.nih.gov/protein/P59594/], residues 1–1190), and SHC014 (AGZ48806.1 [https://www.ncbi.nlm.nih.gov/protein/AGZ48806.1/], residues 1–1191) were created by mutating the S1/S2 cleavage site from RRAR (SARS-CoV-2 and Omicron variants), SLLR (SARS-CoV-1) or SSLR (SHC014) to GSAS and by introducing six proline mutations based on the HexaPro SARS-CoV-2 S (Table S1). DNA segments encoding each HexaPro prefusion-stabilized ectodomain together with a C-terminal T4 fibritin trimerization motif, AviTag, and a his-tag were cloned into pcDNA3.1 between the NcoI and XhoI restriction sites by Gene Universal Inc. (Newark, DE). Expi293F cells (Thermo Fisher Scientific) were transfected with these plasmids using the ExpiFectamine Transfection Kit using the manufacturer's protocol (Thermo Fisher Scientific). Six days after transfection, the cells were centrifuged at 6000 x $g$ for 15 min, and the supernatant was removed. Then, the supernatant was dialyzed into PBS overnight and loaded onto 1 mL of HisPure Ni-NTA resin (Thermo Fisher Scientific) in a gravity flow column (G-Biosciences). The column was washed with 90 mL of binding buffer (150 mM Tris, 150 mM NaCl, 20 mM Imidazole, pH 8). Three mL of elution buffer (150 mM Tris, 150 mM NaCl, 400 mM Imidazole, pH 8) were added to the resin and allowed to incubate for 5 min. The elution buffer was then allowed to flow out of the column, and the elution process was repeated two additional times for a total eluate volume of 9 mL. The eluate was concentrated with a 10 kDa MWCO spin filter (Millipore Sigma), then purified on a Superdex 200 Increase 10/30 column in 20 mM Tris,

20 mM NaCl, pH 8 buffer. Fractions containing S protein were concentrated again and quantified using the bicinchoninic acid assay (BCA) assay (Thermo Scientific).

## Expression and purification of MS2

The DNA encoding the single chain MS2 coat protein dimer was cloned into pET-28b between the NdeI and XhoI restriction sites, and an Avi-Tag was inserted between residues 14 and 15 of the second monomer by GenScript Biotech Corporation (Piscataway, NJ)[14,29]. The MS2-AviTag plasmid was co-transformed with a plasmid containing BirA biotin-protein ligase into BL21(DE3) competent E. coli (New England Biolabs) according to the manufacturer's protocol. The transformation was added to 5 mL of 2xYT media and grown overnight at 37 °C. The 5 mL starter culture was added to 1 L of 2xYT media and incubated at 37 °C and 225 rpm until the optical density reached 0.6. The culture was induced with 1 mM IPTG (Fisher BioReagents). At the same time, D-biotin (50 µM) was added, and the incubator temperature was reduced to 30 °C. The culture was incubated overnight then centrifuged at 7000 x $g$ for 7 min to pellet the cells. The supernatant was decanted, and the pellet was then resuspended in lysis buffer containing 20 mM Tris base (pH 8), lysozyme (0.5 mg/mL), benzonase (125 units; EMD Millipore), and a quarter of a SigmaFast EDTA-free protease inhibitor cocktail tablet (Sigma Aldrich). The resuspended pellet was incubated on ice with occasional swirling for 20 min, after which sodium deoxycholate (Alfa Aesar) was added to a final concentration of 0.1% (w/v). The mixture was sonicated for 3 min at 35% amplitude with a pulse of 3 s on and 3 s off (Sonifier S-450, Branson Ultrasonics). The lysate was cooled on ice for 5 min, then sonication was repeated. The lysed cells were centrifuged at 19,000 x g for 30 min. The supernatant was collected, centrifuged again at 19,000 x g for 20 min, then diluted 3-fold in 20 mM Tris Base, pH 8. 25 mL of the lysate was loaded onto four HiScreen Capto Core columns (Cytvia) in series using an Akta Start system. MS2 was purified by washing with 20 mM Tris Base for ~5 column volumes (CVs). The entire flowthrough was collected in 1.8 mL fractions, and SDS-PAGE was used to determine purity and yield. Fractions 7–12 were typically those most enriched in MS2 while smaller impurities would concentrate in later fractions due to the CaptoCore system's size-exclusion character. Fractions containing MS2 were pooled and concentrated to 1 mL with a 10 kDa MWCO spin filter (Millipore Sigma). MS2 was then further purified by SEC using a Superdex 200 Increase 10/30 column (Cytvia) running in an aqueous buffer (20 mM Tris, 20 mM NaCl, pH 8). Fractions from SEC were pooled and quantified by BCA assay (Thermo Scientific).

## In vitro biotinylation of AviTagged MS2 and S proteins

MS2 and S proteins were biotinylated in vitro with a BirA biotin-protein ligase standard reaction kit (Avidity LLC). Following buffer exchange into 20 mM Tris, 20 mM NaCl, pH 8 buffer, the proteins were concentrated to 45 µM, then BirA and a mixture of biotin, ATP, and magnesium acetate (Biomix B) was added to the protein solution. The solution was allowed to mix overnight at 4 °C. More Biomix B was added, and the solution was mixed at 37 °C for 2 h, followed by the addition of more Biomix B and another overnight incubation at 4 °C. The protein was then purified on a Superdex 200 Increase 10/300 column (Cytiva) to remove BirA and excess biotin and quantified by a BCA assay.

## Expression, refolding, and purification of streptavidin

Streptavidin (SA) was expressed, refolded, and purified essentially as previously described[14,29,52,53]. Briefly, DNA encoding SA (Addgene plasmid #46367, a gift from Mark Howarth) was transformed into BL21(DE3) cells (New England Biolabs) according to the manufacturer's protocol. The transformation was split between four culture tubes containing 5 mL of 2xYT media and grown overnight at 37 °C. Each 5 mL culture was added to 1 L of 2xYT media and incubated at 37 °C

until an OD of 0.6–1.0 was reached. IPTG (Fisher BioReagents) was added to a final concentration of 1 mM to induce expression, and the temperature was reduced to 30 °C. After overnight induction, the cultures were centrifuged at 7000 x g for 7 min to produce two cell pellets. Each pellet was resuspended in 50 mL of lysis buffer (50 mM Tris, 100 mM NaCl, pH 8.0) containing 1 mg/mL lysozyme (Alfa Aesar) and benzonase (500 units; EMD Millipore). The mixture was incubated with mixing for 1 h at 4 °C then homogenized. Sodium deoxycholate (Alfa Aesar) was added to a final concentration of 0.1% (w/v), then the mixture was sonicated for 3 min at 35% amplitude with a pulse of 3 s on and 3 s off. The lysate was centrifuged at 27,000 x g for 15 min, and the supernatant was removed. The pellets were again resuspended in 50 mL of lysis buffer containing 1 mg/mL lysozyme (Alfa Aesar), incubated for 30 min at 4 °C, homogenized, and sonicated. Following centrifugation, the two inclusion body pellets were washed. Each pellet was resuspended in 50 mL of wash buffer #1 (50 mM Tris, 100 mM NaCl, 100 mM EDTA, 0.5% (v/v) Triton X-100, pH 8.0), then homogenized and sonicated at 35% amplitude for 30 s. The lysate was centrifuged at 27,000 x g for 15 min, and the supernatant was discarded. This wash procedure was repeated twice for a total of 3 washes. Both pellets were then resuspended in 50 mL of wash buffer #2 (50 mM Tris, 10 mM EDTA, pH 8.0), homogenized, and sonicated at 35% amplitude for 30 s. The mixture was centrifuged at 15,000 × g for 15 min, and the supernatant was discarded. This process was repeated once more. The inclusion body pellets were then resuspended in 10 mL of resuspension buffer, and guanidine hydrochloride was added to a final concentration of 7.12 M. The mixture was stirred for 1 h at room temperature, then centrifuged for 12 min at 12,000 × g. The supernatant was transferred to a syringe and loaded onto a syringe pump, then the supernatant was added at a rate of 30 mL/h to 1 L of chilled PBS that was stirring rapidly. The solution was stirred overnight at 4 °C, then insoluble protein was pelleted out by centrifuging at 17,000 × g for 15 min. The supernatant was filtered with a 0.45-µm bottle-top filter then stirred vigorously. While stirring, ammonium sulfate was slowly added to the filtrate until a final concentration of 1.9 M was reached. At this point, protein impurities precipitate out. The solution was allowed to mix for 3 h at 4 °C then centrifuged for 15 min at 17,000 × g to remove the precipitate. The solution was filtered with a 0.45-µm bottle-top filter to further remove precipitate. While mixing, ammonium sulfate was added to bring the concentration to 3.68 M and precipitate SA. The mixture was stirred overnight at 4 °C, then the SA was pelleted by centrifuging at 17,000 × g for 15 min. The SA pellet was resuspended in 20 mL of Iminobiotin Affinity Chromatography (IBAC) binding buffer (50 mM Sodium Borate, 300 mM NaCl, pH 11.0). Five mL of Pierce Iminobiotin Agarose (Thermo Scientific) in a gravity flow column (G-Biosciences) was equilibrated with 5 column volumes of IBAC binding buffer, then the SA solution was poured over the resin. 20 column volumes of IBAC binding buffer were added to the column to wash the resin and bound SA. To elute the protein, 8 column volumes of IBAC elution buffer (20 mM Potassium Phosphate, pH 2.2) were added to the column. The eluate was dialyzed into PBS and concentrated using a 10 kDa MWCO centrifugal filter (Millipore Sigma). SA concentration was measured using UV absorbance at 280 nm.

## Assembly and purification of MS2-SA VLPs

A concentrated solution of 20x molar excess SA was stirred vigorously in a 5 mL glass vial, and biotinylated MS2 was slowly added dropwise to the SA. After mixing, a Superdex 200 Increase 10/300 SEC column was used to separate the excess SA from the MS2-SA VLPs. To quantify the SA bound to the MS2 VLP, small samples were mixed with Nu-PAGE lithium dodecyl sulfate (LDS) sample buffer (Invitrogen) and heated at 90 °C for 30 min. After heating, the samples were loaded onto a polyacrylamide gel along with a series of SA standards with known concentrations determined using the UV absorbance at 280 nm. The intensities of the SA bands from the samples were compared to those

of the standards to determine the concentration of SA. This concentration was then used in determining the stoichiometric ratio of MS2-SA to biotinylated S.

## Preparation of VLP-S

Analytical SEC was used to determine the stoichiometric ratio of MS2-SA and biotinylated S. Mixtures containing 10 μg of biotinylated S protein and varying amounts of MS2-SA were run on SEC, and the ratio containing the lowest amount of MS2-SA without any excess S protein appearing on the chromatogram was chosen. MS2-SA and biotinylated S protein were mixed in this ratio, then the mixture was diluted to a final concentration of 0.12 μg S protein/μL. The VLP-S were then characterized by ELISA, DLS, SEC, and TEM.

## Expression of ACE2-Fc and S-binding Antibodies

The variable region of the heavy and light chains of CR3022[54], S309[55], and S2P6[56] were cloned into the TGEX-HC and TGEX-LC vectors (Antibody Design Labs), respectively, according to the manufacturer's protocol. ACE2 (residues 1–615) was cloned into TGEX-HC. The plasmids were expressed in Expi293F cells (ThermoFisher Scientific) using the ExpiFectamine Transfection Kit (Thermo Fisher Scientific) and transfected according to the manufacturer's protocol. After incubation for 6 days at 37 °C for, the cells were centrifuged at 6000 × g for 15 min. The supernatant was diluted in MabSelect Binding Buffer (20 mM sodium phosphate, 150 mM NaCl, pH 7.2) and passed through a 1-mL MabSelect SuRe column (Cytiva) connected to an ÄKTA Start. The column was washed, and the proteins were eluted according to the manufacturer's protocol. Fractions containing the protein were collected and dialyzed into PBS, then the concentration of antibodies and ACE2-Fc was measured using the BCA assay.

## Characterization of S and VLP-S by ELISA

VLP-S and S protein were diluted in PBS such that the concentration of S protein in each solution was 1 μg/mL. 100 μL (0.1 μg of S protein) per well of the diluted protein was then coated onto a Nunc Maxisorp 96-well plate. After a 1-h incubation, the protein solutions were discarded, and the wells were blocked with 200 μL of 5% BSA (EMD Millipore) in PBST (0.05% Tween-20) for 1 h. The wells were then washed three times with PBST. Stock solutions of primary antibodies at 0.3 mg/mL, 3.4 mg/mL, 3.8 mg/mL, and 1.9 mg/mL (by BCA) for ACE2-Fc, CR3022, S309, and S2P6 respectively were diluted 9:1500, 1:1500, 1:1500, and 1:30,000 respectively in PBST with 1% BSA, and 100 μL per well of these diluted primary antibodies were added. After 1 h, the wells were washed three times with PBST, and 100 μL of horseradish peroxidase-conjugated anti-human IgG Fc goat antibody (MP Biomedical, catalog #674171, dilution 1:5000) diluted in PBST with 1% BSA was added to all wells. The wells were washed three times with PBST after a 1-h incubation. 100 μL of TMB (Thermo Scientific) were added to each well and allowed to develop for 3 min. The reaction was stopped with 160 mM sulfuric acid, and the absorbance at 450 nm was read with a Synergy H4 plate reader (BioTek) and Gen5 2.07 software (BioTek).

## SDS-PAGE

Protein samples were diluted in Nu-PAGE lithium dodecyl sulfate (LDS) sample buffer (Invitrogen) and heated for 30 min at 90 °C. 15 μL of protein samples and 2 μL of PageRuler Plus Prestained Protein Ladder were pipetted into the well of a 4–12% Bi-Tris gel (Invitrogen). Gels were then run in MES-SDS buffer at 4 °C for 60 min at 110 V. Afterwards, the gel was washed in DI water and stained with Imperial Protein Stain for 30 min then destained overnight. Gels were then imaged with ChemiDoc MP imaging system and Image Lab 5.2.1 software (Bio-Rad).

## Analytical SEC

SEC was run using a Superdex 200 Increase 10/300 Column (Cytiva) and Unicorn 7 control system (Cytiva). 950 μL samples of either S protein or VLP-S, each containing 10 μg of S, were injected into the sample loop. The loop was flushed with PBS to inject the sample onto the column, then the protein was eluted with 1 column volume of PBS flowing at 0.5 mL/min. During elution, absorbance at 280, 210, and 205 nm was monitored.

## Dynamic light scattering

VLP-S was diluted in PBS to a concentration of 0.05 μg/μL, added to a UVette (Eppendorf), and inserted into a DynaPro NanoStar Dynamic Light Scattering detector (Wyatt Technology). Ten acquisitions per sample were collected at 25 °C and displayed as % Mass with the Isotropic Spheres model.

## Transmission electron microscopy

Conventional negative-stain transmission electron microscopy (NS-TEM) was performed on VLP-S variants, as described previously[14,29]. Briefly, 4 μl of the diluted samples were applied onto glow-discharged 200 mesh copper grids (CF200-Cu; Electron Microscopy Sciences, PA), washed with distilled water (3x) and stained in droplets of 1% phosphotungstic acid (PTA, pH 6–7) for 1 min. The grids were drained from the grid backside and then air-dried. The stained grids were imaged with a low dose of 50–60 e⁻/Å², under a nominal magnification of 73 kx (pixel size of 2.0 Å), defocus of −0.5 to −2 μm, on a Talos L120C transmission electron microscope (ThermoFisher Scientific, Hillsboro, OR), operating at 120 kV. Images were acquired on a 4 K x 4 K Ceta CMOS camera (ThermoFisher Scientific, Hillsboro, OR), using SerialEM 3.84[57].

## Amino acid identity and phylogenetic trees

Amino acid sequences for spike proteins from all included sarbecoviruses were retrieved from GenBank (Table S2). Sequences were aligned using Clustal Omega 1.2.4 (Conway Institute, UCD Dublin), and the percentage of identical amino acids for each pair was calculated from this alignment. An RBD alignment was generated by copying the full S alignment from residues 319 through 541 inclusive of SARS-CoV-2 614D (YP_009724390.1). Maximum likelihood phylogenetic trees were generated from the RBD alignment using PhyML 3.3.20220408 (Stephane Guindon, University of Montpellier) using the LG amino acid substitution model and a maximum parsimony starting tree. Phylogenetic trees were visualized using TreeViewer 2.0.1 (Giorgio Bianchini, University of Bristol).

## Cells and Virus

Virus stocks were generated on Vero E6 TMPRSS2 cells obtained from the National Institute of Infectious Diseases, Japan[58] which were maintained in high glucose Dulbecco's modified Eagle's medium (DMEM) containing 10% fetal bovine serum (FBS) and antibiotic/antimycotic solution along with G418 (1 mg/ml). Tissue titrations were performed on Vero E6 TMPRSS2-T2A-ACE2 cells obtained from Dr. Barney Graham, NIAID Vaccine Research Center which were maintained in DMEM supplemented with 10% FBS, 10 mM HEPES (pH 7.3) and antibiotic/antimycotic solution along with puromycin (10 μg/ml) The following viruses, a prototypical ancestral isolate (SARS-CoV-2/UT-HP095-1 N/Human/2020/Tokyo), BA.1 isolate (hCoV-19/USA/WI-WSLH-221686/2021), BA.5 isolate (hCoV-19/Japan/TY41-702/2022), XBB.1 isolate (hCoV-19/USA/NY-MSHSPSP-PV73997/2022), and two recombinant SARS-like bat CoVs, SHC014 and WIV1, used in studies were provided by Ralph Baric[32,59].

## Animal experiments and approval

Animal studies were performed under a protocol approved by the Institutional Animal Care and Use Committee at the University of Wisconsin, Madison (protocol number V006426). Virus infections were performed under isoflurane, and all efforts were made to minimize pain. Animal studies were not blinded. Group sizes were

determined based on prior virus challenge studies, and no sample-size calculations were performed to determine the power of each study.

## Experimental infection of syrian hamsters

Female Syrian wild-type hamsters (4–5 weeks old, Envigo) or K18-human ACE2 homozygous transgenic hamsters[60] from an established colony at UW-Madison (females, 5-6 weeks old) were used in this study. Hamsters were vaccinated with the indicated vaccine candidate (15 μg of each S protein) with adjuvant (alhydrogel, 4.5–5.5 mg; equal volume of vaccine and adjuvant) by subcutaneous inoculation. A separate group of animals was vaccinated by intramuscular inoculation in the thigh muscle with 30 μg of Pfizer-BioNTech's bivalent vaccine (residual material stored at −80 °C 24 h or less after the vaccine was reconstituted at a university health clinic). Under isoflurane anesthesia, animals were infected by intranasal inoculation with the indicated virus isolates at a dose of $10^5$ plaque-forming units (pfu) in 30 μl of total volume. Three days after infection, animals were humanely sacrificed by overdose of isoflurane and lung tissue and nasal turbinate samples were collected to measure the amount of virus.

## Focus reduction neutralization test (FRNT)

Neutralization of all viruses was characterized by using a focus reduction neutralization test. Serial dilutions of serum from vaccinated hamsters starting at a final concentration of 1:20 were mixed with ~1000 focus-forming units (FFU) of the indicated virus/well and incubated for 1 h at 37 °C. Pooled serum from hamsters vaccinated with VLP without the S protein served as a control. The antibody-virus mixture was inoculated onto Vero E6/TMPRSS2 cells in 96-well plates and incubated for 1 h at 37 °C. An equal volume of methylcellulose solution was added to each well. The cells were incubated for 16 h at 37 °C and then fixed with formalin. After the formalin was removed, the cells were immunostained with a mouse monoclonal antibody against SARS-CoV-1/2 nucleoprotein (clone 1C7C7, Sigma-Aldrich, catalog #MA5-29982, 1:10,000 dilution), followed by a horseradish peroxidase-labeled goat anti-mouse immunoglobulin (ThermoFisher, catalog #31430, 1:2000 dilution). The infected cells were stained with TrueBlue Substrate (SeraCare Life Sciences) and then washed with distilled water. After cell drying, the focus numbers were quantified by using an ImmunoSpot S6 Analyzer, ImmunoCapture software, and BioSpot software (Cellular Technology). The results are expressed as the 50% focus reduction neutralization titer (FRNT50). The FRNT50 values were calculated by using Prism 9 (Graphpad Software). Percent neutralization was calculated as $N = 100\% \times (1 - \frac{F_v}{F_c})$, where $N$ is the percent neutralization, $F_v$ is the number of foci in the presence of sera from hamsters vaccinated with VLP-S, and $F_c$ is the number of foci in the presence of pooled sera from hamsters vaccinated with VLP control. The FRNT50 value was then calculated from the normalized percent neutralization using a four-parameter nonlinear regression in Graphpad Prism.

## Antigenic cartography

Antigenic cartography was performed using the Racmacs R package (Racmacs 1.1.35 with R 4.2.1 and RStudio 2022.07.01)[61]. In brief, FRNT50 values were calculated from each serum sample against each virus. The dissimilarity $D_{ij}$ between each serum $i$ and virus $j$ is defined as $D_{ij} = -\log_2 H_{ij} + \log_2 H_{i,\max}$ where $H_{ij}$ is the FRNT50 value of serum $i$ against virus $j$ and $H_{i,\max}$ is the maximum FRNT50 value from serum $i$. The error function for each pair is then $E_{ij} = (D_{ij} - d_{ij})^2$, where $d_{ij}$ is the Euclidean distance on the two-dimensional map between serum $i$ and virus $j$. For FRNT50 values below a detection threshold (e.g., <20), the error function was instead calculated as $E_{ij} = \frac{a^2}{1 + e^{-10a}}$ for $a = D_{ij} - 1 - d_{ij}$. The summed error function was then minimized using conjugate gradient descent optimization, and 5000 restarts with random starting positions were used to approximate the global optimum.

## Biosafety statement

Research with SARS-CoV-2 and related SARS-like viruses was performed under biosafety level 3 agriculture (BSL-3Ag) containment at the Influenza Research Institute with an approved protocol reviewed by approved the University of Wisconsin-Madison's Institutional Biosafety Committee. The laboratory is designed to meet and exceed the standards outlined in Biosafety in Microbiological and Biomedical Laboratories (6th edition).

## Statistics and reproducibility

SDS-PAGE gels of S and VLP-S (Fig. 2a, c) were conducted twice from the same preparation for each sample with similar results. For TEM imaging (Fig. 2f), at least 70 images were collected and analyzed from one VLP-S preparation per sample with similar results. Characterization of the binding of ACE2-Fc and S2P6 by ELISA (Fig. 2e) was conducted once with three technical replicates for each condition. Data are presented as the mean ± SD. For in vivo characterization of monovalent VLP-S (Fig. 3), there were seven groups with three hamsters each. Neutralization titers (Figs. 3a and 4a) were measured by focus reduction neutralization test (FRNT) conducted as a single assay using sera from each hamster. Viral titers in the lungs and nasal turbinates of Syrian hamsters immunized with VLP-S vaccines (Figs. 3b and 4b, d) 3 days after infection with BA.5 and XBB.1 variants were presented as the mean ± SD ($n = 6$ for BA.5 VLP-control, $n = 3$ for all other groups). The significance was determined by a one-way analysis of variance (ANOVA) and Dunnett post-hoc multiple comparison between groups ($α = 0.05$) for BA.5 variant lung and nasal titers for monotypic vaccines and by Brown-Forsythe and Welch ANOVA tests and Dunnett's T3 multiple comparisons between groups for BA.5 variant and XBB.1 variant lung and nasal titers for cocktail vaccines. Viral titers in the lungs and nasal turbinates of hamsters immunized with Pfizer-BioNTech bivalent vaccine (Fig. 4c, e) 3 days after infection with BA.5 and XBB.1 variants were presented as the mean ± SD ($n = 4$). The significance was determined by a two-tailed unpaired Student's $t$ test. Viral titers in the lungs and nasal turbinates of hACE2 hamsters (Fig. 5) 3 days after infection with WIV1 and SHC014 viruses were presented as the mean ± SD ($n = 4$). Significance was determined by two-tailed unpaired Welch's $t$ tests for lung titers and two-tailed unpaired Student's $t$ tests for nasal titers. For all tests of significance, assumptions of the normality of residuals and homogeneity of variance were validated by the D'Agostino-Pearson test and the Brown-Forsythe test, respectively. All statistical analysis was carried out using Prism 9 (GraphPad).

## Reporting summary

Further information on research design is available in the Nature Portfolio Reporting Summary linked to this article.

# Data availability

All data supporting the conclusions of this paper can be found within the paper, Supplementary Information, and Source Data file. Protein sequences for MS2-AviTag and the HexaPro S proteins are available in Supplementary Table 1. GenBank and RefSeq accession numbers for Fig. 1 are available in Supplementary Table 2. Structures used to generate Fig. 2b are available from the PDB using accession codes 2MS2, 3RY2, and 6VSB. Unprocessed SDS-PAGE gel images for Fig. 2 are available in Supplementary Fig. 3. Source data for Fig. 2e; Supplementary Fig. 2; Figs. 3a–c; 4; and 5 are available in the Source Data file. Source data are provided with this paper.

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

## Acknowledgements

Addgene plasmid #46367 (pET21-Streptavidin-Glutamate_Tag) was a gift from Mark Howarth. RSK acknowledges support from the Garry Betty/ V Foundation Chair Fund at the Georgia Institute of Technology. This work was supported in part by the Department of Biochemistry at the University of Wisconsin, Madison. We are grateful for the use of facilities and instrumentation at the Cryo-EM Research Center in the Department of Biochemistry at the University of Wisconsin, Madison. Y.K. and R.S.K. acknowledge support by the National Institute of Allergy and Infectious Diseases of the National Institutes of Health under Award Number P01 AI165077. Y.K. also acknowledges support from a Research Program on Emerging and Reemerging Infectious Diseases (JP21fk0108552 and JP21fk0108615), a Project Promoting Support for Drug Discovery (JP21nf0101632), the Japan Program for Infectious Diseases Research and Infrastructure (JP22wm0125002), and the University of Tokyo Pandemic Preparedness, Infection and Advanced Research Center (UTOPIA) grant (JP223fa627001) from the Japan Agency for Medical Research and Development. This material is based upon work (by K.L.) supported by the National Science Foundation Graduate Research Fellowship Program under Grant No. DGE-2039655NSF. Any opinions, findings, and conclusions or recommendations expressed in this material are those of the authors and do not necessarily reflect the views of the National Science Foundation. The content is solely the responsibility of the authors and does not necessarily represent the official views of the National Institutes of Health.

## Author contributions

All authors: P.J.H., K.L., A.D., M.K., J.E.Y., E.R.W., Y.K. and R.S.K. were involved in the study design. P.J.H., K.L., A.D., M.K. and J.E.Y. performed data collection and data analysis. P.J.H., K.L., A.D. and R.S.K. wrote the manuscript. All authors read and approved the final manuscript.

## Competing interests

The authors declare the following competing interests: Y.K. has received unrelated funding support from Daiichi Sankyo Pharmaceutical, Toyama Chemical, Tauns Laboratories, Inc., Shionogi & Co. LTD, Otsuka Pharmaceutical, KM Biologics, Kyoritsu Seiyaku, Shinya Corporation, and Fuji Rebio. The other authors declare no competing interests.
