## [Peer Review File · Nature Communications]

Broad Protection Against Clade 1 Sarbecoviruses After a Single Immunization with Cocktail Spike-Protein-Nanoparticle VaccineReviewers' Comments:

Reviewer #1:

Remarks to the Author:

This manuscript describes a novel multi-valent nanoparticle vaccine and shows that it confers protection in a hamster challenge. The vaccine is based off a platform of streptavidin coated VLPs that can combine with biotinylated spike proteins to assemble into spike-coated MS2 VLPs. Trivalent cocktails of these nanoparticles were examined in different challenge models. Overall the manuscript is well written and conveys the goals and successes of the research well. There are limitations inherent in the manuscript – only showing lung titers and not any other sublethal measures in the studies – but it is challenge with omicron variants in hamsters. I think it is suitable for publication in Nat. Comms. Following minor revision, including some important nanoparticle characterization experiments.

Major Comments:

1. While some VLP platforms are not amenable to mosaics, it is confusing why the authors did not consider a mosaic vaccine with this platform. Perhaps in the discussion the reasoning could be explained. If it was tested and worse, that result should be added as it would be incredibly interesting to the field.

2. I agree that the “diameter was measured to be 90 – 100 nm, consistent with prior characterization,” according to the previous paper:

Chiba, S., Frey, S.J., Halfmann, P.J. et al. Multivalent nanoparticle-based vaccines protect hamsters against SARS-CoV-2 after a single immunization. *Commun Biol* 4, 597 (2021).

<https://doi.org/10.1038/s42003-021-02128-8>

But I would consider an additional analysis to confirm particle integrity. Perhaps TEM, as was done previously. This could help rule out that the spike component on some fraction was unfolded. The limited data – DLS and ELISA, are both crude methods, particularly the way the ELISA was done, so in my opinion this is the aspect of the manuscript that needs the most improvement.

3. The authors could do a better job citing other COVID nanoparticle vaccines, which seems like an oversight given the platform. Most of the introduction is spent talking in length about current bivalent vaccines, but if these were efficacious we wouldn't need this manuscript. I would consider shortening this section and describing, to some extent, nanoparticle COVID vaccines. Potential additional references:

a. M. Gordon Joyce et al., vA SARS-CoV-2 ferritin nanoparticle vaccine elicits protective immune responses in nonhuman primates. *Sci. Transl. Med.* 14, eabi5735(2022). DOI: 10.1126/scitranslmed.abi5735

b. Weidenbacher, P.A.B., Sanyal, M., Friedland, N. et al. A ferritin-based COVID-19 nanoparticle vaccine that elicits robust, durable, broad-spectrum neutralizing antisera in non-human primates. *Nat Commun* 14, 2149 (2023). <https://doi.org/10.1038/s41467-023-37417-9>

c. Brouwer PJM, Et. al, Two-component spike nanoparticle vaccine protects macaques from SARS-CoV-2 infection. *Cell*. 2021 Mar 4;184(5):1188-1200.e19. doi: 10.1016/j.cell.2021.01.035. Epub 2021 Jan 26. PMID: 33577765; PMCID: PMC7834972.

d. Etc.

4. I find the Pfizer comparator to be not informative and confusing. Without a control indicating this vaccine retained potency, it's difficult to understand why it shows no detectable efficacy? There is language in the text hypothesizing what happened, but if this serum is available, I would test it for binding (hopefully more sensitive than neutralization) to demonstrate the animals were properly immunized – or provide a citation that a 10ug dose doesn't confer protection, neutralization, or any apparent activity in hamsters. In Fig 4c there is protection with no neutralization – so to not show any protection suggests, in my mind, the procured vaccine was ineffective.

5. The result that “The bivalent vaccine provided complete protection against the XBB.1 challenge despite eliciting significantly lower neutralizing antibody titers against XBB.1 than the trivalent vaccines.” Should be further discussed. Does this data argue against the inclusion of an omicron variant? Does this suggest neutralization is a bad surrogate for protection?

6. Given how similar all the ELISAs look, negative controls should be shown for the ELISAs – ideally the VLP alone binding to Ab control.

Minor Comments:

- Please use fewer significant digits in Fig. 3 top ($278616 = 2.8 \times 10^5$ or $2.8e5$)
- Please provide the dose of aluminum in the alhydrogel adjuvanted samples (as opposed to “equal volume”).
- I like the analysis in Figure 3c
- “While a naïve animal might respond equally to all components of a vaccine, an animal with pre-existing immunity might instead be biased towards one component to which it has been exposed to before.” – please add a citation for original antigenic sin/immunological imprinting.
- Methods around “HiScreen Capto Core” need elaboration. My understanding was that these columns did not bind to large nanoparticles, so the discussion around elution in just 20mM tris is surprising? Did you collect the entire flow through?
- ELISA methods should include coating concentration and Ab concentration – methods as written could not be replicated. “The final concentration of ACE2-Fc and antibodies was such that there would be one molecule per S protein trimer.” Is confusing, and assumes 100% bound to the plate? But we know the vast majority of protein remains unbound to the ELISA plate.

Reviewer #3:

Remarks to the Author:

The study by Halfmann et al evaluates the use of a Spike based nanoparticle vaccine to produce broad protection against Sarbecoviruses. This model uses VLP composed of hexapropylamine Spike proteins linked to the MS2 phage protein to produce a nanoparticle. This vaccine had previously been used in a variety of studies based on the D614G Spike sequence. In this study, the authors have tested the immunogenicity and cross neutralization capacity of a range of Sarbecovirus Spike proteins to identify the minimal combination that can be used together and still produce broadened neutralization and protection in a hamster model of coronavirus infection.

The authors show that Spike protein nanoparticles from a variety of SARS2 viruses, SARS1 and SHC014 are able to neutralize some but not all of the matched strains. They then chose combination of either D614D/SHC014 or 2 other combinations adding in a third Spike protein from BA.2.75.2 or XBB. In hamster vaccination models, the combination of D614G/SHC/XBB produced broadly neutralizing antibody and protection in WT hamsters challenged with XBB.1 and in hACE2 transgenic hamsters challenged with WIV1 or SHC014. No virus was detected in lungs of either of these challenged hamsters demonstrating robust protection from a single vaccine dose. As a control the Pfizer/Biontech vaccine was used at a single dose of 19ug, and it failed to produce neutralizing antibody as well as protect in hamsters. The study is well performed and the addition of the hACE2 hamster models add to the robustness of the data to show broader protection from challenge.

Questions about the study are below

1. While not needed in this manuscript, I would be interested to know if lower protein concentrations have been tested in other hamster vaccine experiments since submission. 15ug of each nanoparticle (45ug in all) is a large dose. The data suggest that much lower concentrations should be as effective. Have then been tested?

2. Mucosal immunity is a major driver of protective responses in humans, and while not as dramatic in hamster models, it would be intriguing to know if neutralizing antibody levels were tested in the vaccinated hamsters. Also were nasal washes analyzed for virus titer after challenge?

Several interesting points are broad up by the data that should be commented on in the discussion:

1. Is there a structural basis for SHC014 Spike to neutralize better against WIV1 than SARS1? Has that been seen in other vaccine models or is this unique to this nanoparticle design?
2. A separate discussion is warranted in this manuscript for explaining implications of this work across the context of broadly protective vaccines. Important points to touch on are 1) mucosal vs systemic immunity of this vaccine platform, 2) structural basis for the neutralization and protection with emphasis on other widely divergent bat coronaviruses and how they may fair against this vaccine based on other published structural data, 3) longevity of the response and how this vaccine design could be used commercially.

Response to Reviewers:

Reviewer #1

Reviewer 1 noted that *“Overall the manuscript is well written and conveys the goals and successes of the research well. There are limitations inherent in the manuscript – only showing lung titers and not any other sublethal measures in the studies – but it is challenge with omicron variants in hamsters. I think it is suitable for publication in Nat. Comms. Following minor revision, including some important nanoparticle characterization experiments.”*

We thank the reviewer and have responded to the other comments below. We have also added nasal turbinate virus titers as an additional sublethal measure in response to the reviewer’s comment.

Comment: *While some VLP platforms are not amenable to mosaics, it is confusing why the authors did not consider a mosaic vaccine with this platform. Perhaps in the discussion the reasoning could be explained. If it was tested and worse, that result should be added as it would be incredibly interesting to the field.*

Response: We did not test mosaic vaccines as part of the current work and have added the following text to the discussion to explain the reasoning as suggested by the reviewer:

“We did not test mosaic vaccines as part of the current work. While the underlying concept is interesting, a recent study reported a head-to-head comparison between cocktail and mosaic approaches for spike protein-nanoparticle vaccines and did not identify a significant difference in neutralization titers between the two approaches.²⁴ A cocktail vaccine is also aligned with the hypothesis motivating this work – that if the mixture of antigens comprising the cocktail is carefully selected by characterizing the antigenic landscape, the antibody response elicited by each component would collectively result in a broadly protective polyclonal antibody response. It would be interesting to revisit the mosaic idea in future work.”

Comment: *I agree that the “diameter was measured to be 90 – 100 nm, consistent with prior characterization,” according to the previous paper:*

Chiba, S., Frey, S.J., Halfmann, P.J. et al. Multivalent nanoparticle-based vaccines protect hamsters against SARS-CoV-2 after a single immunization. Commun Biol 4, 597 (2021). <https://doi.org/10.1038/s42003-021-02128-8>

But I would consider an additional analysis to confirm particle integrity. Perhaps TEM, as was done previously. This could help rule out that the spike component on some fraction was unfolded. The limited data – DLS and ELISA, are both crude methods, particularly the way the ELISA was done, so in my opinion this is the aspect of the manuscript that needs the most improvement.

Response: We have added TEM characterization as requested by the reviewer as Figure 2f in the revised manuscript.

Comment: *The authors could do a better job citing other COVID nanoparticle vaccines, which seems like an oversight given the platform. Most of the introduction is spent talking in length about current bivalent vaccines, but if these were efficacious we wouldn’t need this manuscript. I would consider shortening this section and describing, to some extent, nanoparticle COVID vaccines. Potential additional references:*

a. M. Gordon Joyce et al., A SARS-CoV-2 ferritin nanoparticle vaccine elicits protective immune responses in nonhuman primates. *Sci. Transl. Med.* 14, eabi5735 (2022).

DOI:10.1126/scitranslmed.abi5735

b. Weidenbacher, P.A.B., Sanyal, M., Friedland, N. et al. A ferritin-based COVID-19 nanoparticle vaccine that elicits robust, durable, broad-spectrum neutralizing antisera in non-human primates. *Nat Commun* 14, 2149 (2023). <https://doi.org/10.1038/s41467-023-37417-9>

c. Brouwer PJM, Et. al, Two-component spike nanoparticle vaccine protects macaques from SARS-CoV-2 infection. *Cell.* 2021 Mar 4;184(5):1188-1200.e19. doi: 10.1016/j.cell.2021.01.035. Epub 2021 Jan 26. PMID: 33577765; PMCID: PMC7834972.

Response:

We have cited these manuscripts and added the following text to the introduction based on the reviewer's suggestions:

"Protein nanoparticles have emerged as attractive platforms for the display of S protein antigens. Brouwer et al. generated two component protein nanoparticles displaying stabilized prefusion SARS-CoV-2 S proteins that protected vaccinated macaques from a challenge with SARS-CoV-2.²⁵ Joyce et al. showed that adjuvanted SARS-CoV-2 S protein-ferritin nanoparticle vaccines protected non-human primates from a challenge with SARS-CoV-2.²⁶ Weidenbacher et al. reported that adjuvanted ferritin nanoparticle vaccines displaying a truncated form of the SARS-CoV-2 S protein ectodomain elicited a broad neutralizing antibody response in non-human primates.²⁷ Hutchinson et al. designed self-assembling protein nanoparticles displaying multiple S protein antigens that protected mice from a challenge with MERS-CoV.²⁸ As described above, Brinkemper et al. designed nanoparticles presenting mixtures of the SARS-CoV-1 and SARS-CoV-2 S proteins.^{24"}

Comment: *I find the Pfizer comparator to be not informative and confusing. Without a control indicating this vaccine retained potency, it's difficult to understand why it shows no detectable efficacy? There is language in the text hypothesizing what happened, but if this serum is available, I would test it for binding (hopefully more sensitive than neutralization) to demonstrate the animals were properly immunized – or provide a citation that a 10ug dose doesn't confer protection, neutralization, or any apparent activity in hamsters. In Fig 4c there is protection with no neutralization – so to not show any protection suggests, in my mind, the procured vaccine was ineffective.*

Response: We agree with the reviewer's concerns regarding the potency of a 10 µg dose of the Pfizer-BioNTech vaccine and have repeated the experiment with a higher dose (30 µg). This was a sufficient dose to result in a significant decrease in lung titers following an XBB.1 challenge, and the higher dose did elicit neutralizing antibodies against 614G, BA.5, and in one case against XBB.1. We have updated the text to include these results.

Regarding neutralization:

"To compare, hamsters immunized with 30 µg of the Pfizer-BioNTech bivalent vaccine did produce neutralizing antibodies against an early SARS-CoV-2 isolate (S-614G) and BA.5 as expected given the composition of the vaccine. Like our bivalent 614D/SHC014 vaccine, neutralization titers were significantly reduced against XBB.1 (3 out of 4 hamsters had no detectable titers), but unlike any of our bivalent or trivalent vaccines, no detectable neutralization titers were observed against the Clade 1A bat CoVs SHC014 and WIV1. These results further support the inclusion of antigens from both XBB-like and Clade 1A CoVs in a broadly protective cocktail vaccine."

Regarding protection from challenge

“Meanwhile, a single immunization with 30 µg of the Pfizer-BioNTech bivalent vaccine significantly decreased lung and nasal titers after challenge with XBB.1 (Fig. 4c,e).”

Comment: *The result that “The bivalent vaccine provided complete protection against the XBB.1 challenge despite eliciting significantly lower neutralizing antibody titers against XBB.1 than the trivalent vaccines.” Should be further discussed. Does this data argue against the inclusion of an omicron variant? Does this suggest neutralization is a bad surrogate for protection?*

Response: We have added the following discussion to the revised manuscript as suggested by the reviewer:

“While neutralizing antibody titers against SARS-CoV-2 have been identified as a correlate of protection against symptomatic disease^{8,39}, even low neutralizing titers can be sufficient for protection. High neutralization titers elicited by vaccination are still desirable, since they are likely to provide greater protection against viral escape; neutralization titers against viral variants are generally lower than those against the vaccine strain. The inclusion of an Omicron antigen in the cocktail is thus justified, particularly in light of the continuing emergence of new Omicron variants.”

Comment: *Given how similar all the ELISAs look, negative controls should be shown for the ELISAs – ideally the VLP alone binding to Ab control.*

Response: We have added VLP-only controls to Figure 2 and Supplementary Figure 2b.

We have also made changes in response to the reviewer’s other minor comments:

Comment: *Please use fewer significant digits in Fig. 3 top ($278616 = 2.8 \times 10^5$ or $2.8e5$)*

Response: We thank the reviewer for the recommendation and have updated Figure 3 accordingly.

Comment: *Please provide the dose of aluminum in the alhydrogel adjuvanted samples (as opposed to “equal volume”).*

Response: We have added the dose of aluminum in the alhydrogel samples as requested.

Comment: *I like the analysis in Figure 3c*

Response: We thank the reviewer for this encouraging comment.

Comment: *“While a naïve animal might respond equally to all components of a vaccine, an animal with pre-existing immunity might instead be biased towards one component to which it has been exposed to before.” – please add a citation for original antigenic sin/immunological imprinting.*

Response: We have added citations for “original antigenic sin” as suggested by the reviewer.

- Francis Jr., T. On the Doctrine of Original Antigenic Sin *Proc Am Phil Soc* **104**, 572-578 (1960)
- de St. Groth, F. & Webster, R.G. Disquisitions of Original Antigenic Sin. I. Evidence in man. *J Exp Med* **124**, 331-345 (1966)
- de St. Groth, F. & Webster, R.G. Disquisitions on Original Antigenic Sin. II. Proof in lower creatures. *J Exp Med* **124**, 347-361 (1966).

Comment: *Methods around “HiScreen Capto Core” need elaboration. My understanding was that these columns did not bind to large nanoparticles, so the discussion around elution in just 20mM tris is surprising? Did you collect the entire flow through?*

Response: We have updated our description of MS2 purification using the HiScreen Capto Core columns. The revised text reads:

“25 mL of the lysate was loaded onto four HiScreen Capto Core columns (Cytvia) in series using an Akta Start system. MS2 was purified by washing with 20 mM Tris Base for ~5 column volumes (CVs). The entire flowthrough was collected in 1.8 mL fractions, and SDS-PAGE was used to determine purity and yield. Fractions 7-12 were typically those most enriched in MS2 while smaller impurities would concentrate in later fractions due to the CaptoCore system’s size-exclusion character.”

Comment: *ELISA methods should include coating concentration and Ab concentration – methods as written could not be replicated. “The final concentration of ACE2-Fc and antibodies was such that there would be one molecule per S protein trimer.” Is confusing, and assumes 100% bound to the plate? But we know the vast majority of protein remains unbound to the ELISA plate.*

Response: We have rewritten the ELISA methods as follows:

“VLP-S and S protein were diluted in PBS such that the concentration of S protein in each solution was 1 µg/mL. 100 µL (0.1 µg of S protein) per well of the diluted protein was then coated onto a Nunc Maxisorp 96-well plate. After a 1-hour incubation, the protein solutions were discarded, and the wells were blocked with 200 µL of 5% BSA (EMD Millipore) in PBST (0.05% Tween-20) for 1 hour. The wells were then washed three times with PBST. Stock solutions of primary antibodies at 0.3 mg/mL, 3.4 mg/mL, 3.8 mg/mL, and 1.9 mg/mL (by BCA) for ACE2-Fc, CR3022, S309, and S2P6 respectively were diluted 9:1500, 1:1500, 1:1500, and 1:30000 respectively in PBST with 1% BSA, and 100 µL per well of these diluted primary antibodies were added. After 1 hour, the wells were washed three times with PBST, and 100 µL of horseradish peroxidase-conjugated anti-human IgG Fc goat antibody (MP Biomedical, 1:5000) diluted in PBST with 1% BSA was added to all wells. The wells were washed three times with PBST after a 1-hour incubation. 100 µL of TMB (Thermo Scientific) were added to each well and allowed to develop for 3 minutes. The reaction was stopped with 160 mM sulfuric acid, and the absorbance at 450 nm was read with a Spectramax i3x plate reader (Molecular Devices) and Gen5 2.07 software (BioTek).”

Reviewer #2

Reviewer 2 noted that *“The study is well performed and the addition of the hACE2 hamster models add to the robustness of the data to show broader protection from challenge.”*

We thank the reviewer and have responded to the reviewer’s other comments below.

Comment: *While not needed in this manuscript, I would be interested to know if lower protein concentrations have been tested in other hamster vaccine experiments since submission. 15ug of each nanoparticle (45ug in all) is a large dose. The data suggest that much lower concentrations should be as effective. Have then been tested?*

Response: We have not yet tested reducing the dose of our vaccine cocktails. We hypothesize that it would be possible to reduce the dose though, since our trivalent cocktails elicit robust and broad neutralizing antibody titers that are likely to be well above the threshold needed for protection.

Comment: *Mucosal immunity is a major driver of protective responses in humans, and while not as dramatic in hamster models, it would be intriguing to know if neutralizing antibody levels were*

tested in the vaccinated hamsters. Also were nasal washes analyzed for virus titer after challenge?

Response: We have added data for nasal turbinate titers in Figures 3c, 4d-e, and 5b. Discussion of these nasal turbinate titers has also been added.

Regarding nasal titers after monovalent vaccination and challenge with BA.5 (Figure 3c):

“Similar trends were observed in viral titers in the nasal turbinates as VLP-614D-S and all of the Omicron VLP-S vaccines significantly reduced virus levels, while hamsters vaccinated with VLP-SHC014-S and VLP-SARS-CoV-1-S had viral levels comparable to those of the control hamsters (Fig. 3c).”

Regarding nasal titers after cocktail vaccination and challenge with BA.5 and XBB.1 (Figure 4d):

“Similar trends were also seen for nasal titers. The VLP-614D-S and VLP-SHC014-S bivalent cocktail significantly reduced nasal virus titers, while hamsters immunized with either trivalent cocktail had undetectable nasal titers (Fig 4d).”

Regarding nasal titers after Pfizer-BioNTech bivalent vaccination and challenge with XBB.1 (Figure 4e):

“Meanwhile, a single immunization with 30 µg of the Pfizer-BioNTech bivalent vaccine significantly decreased lung and nasal titers after challenge with XBB.1 (Fig. 4c,e).”

Regarding nasal titers after cocktail vaccination in hACE2 hamsters and challenge with WIV1 and SHC014 (Figure 5b):

“Nasal turbinate titers were also significantly reduced in vaccinated transgenic hamsters in both cases (Fig. 5b), though the magnitude of this reduction was weaker than for the Clade 1B viruses.”

Comment: *Several interesting points are brought up by the data that should be commented on in the discussion:*

1. Is there a structural basis for SHC014 Spike to neutralize better against WIV1 than SARS1? Has that been seen in other vaccine models or is this unique to this nanoparticle design?

Response: We thank the reviewer for this comment; as seen in Fig. 1c, the WIV1 S is in fact more homologous to the SHC014 S than to the SARS-CoV-1 S. We have corrected this text in the revised manuscript, which now reads

“VLP-SHC014-S elicited higher neutralizing antibody titers against WIV1 compared to VLP-SARS-CoV-1-S, consistent with the WIV1 S being more homologous to the SHC014 S than to the SARS-CoV-1 S (Fig. 1c).”

Comment: *2. A separate discussion is warranted in this manuscript for explaining implications of this work across the context of broadly protective vaccines. Important points to touch on are 1) mucosal vs systemic immunity of this vaccine platform, 2) structural basis for the neutralization and protection with emphasis on other widely divergent bat coronaviruses and how they may fair against this vaccine based on other published structural data, 3) longevity of the response and how this vaccine design could be used commercially.*

Response: We have added the following text to the discussion based on the reviewer's comments:

“While we are encouraged by these results, there are several additional avenues that would be interesting to explore in future work. Enhancing mucosal immunity might not only enhance protection against viral infection, but also decrease viral transmission.⁴⁰ Intranasal vaccination against SARS-CoV-2 has been explored with several platforms, including mRNA-lipid nanoparticles⁴¹, nanoparticles displaying the RBD⁴², live attenuated influenza viruses also encoding the RBD⁴³, adenovirus-vectored vaccines⁴⁴, and by using an intranasal boost with the unadjuvanted spike protein⁴⁰. The adaptation of our platform for intranasal delivery could be a promising avenue for improving the mucosal response. Characterizing the longevity of protection would also be an interesting avenue for future research. It would be particularly interesting to determine whether stronger mucosal immunity results in more durable protection against symptomatic disease.

Secondly, while the focus of this paper was on protecting against viruses in Clades 1A and 1B, extending protection to further sarbecovirus clades would be interesting to explore. Significant differences in their receptor binding domains are the primary basis for virus classification into clades, with Clade 2 viruses being unable to use ACE2 as an entry receptor and Clade 3 & 4 viruses harboring one deletion relative to Clade 1.⁴⁵ Nevertheless, some Clade 2 viruses (a proposed “Clade 2A”)⁴⁶ and several Clade 3 viruses⁴⁷ may be capable of infecting human cells, albeit in some cases only with exogenous protease treatment. RBD- and NTD-focused humoral immunity is unlikely to be cross-reactive between clades, as inter-clade RBD and NTD amino acid identity percentages are approximately 65-75% and 45-55%, respectively. As such, the inclusion of additional antigens to cover these clades may be necessary. Using the approach described in this work, antigenic cartography should help us identify the minimal mixture of antigens required to elicit broad protection against sarbecoviruses.

Given that protein-nanoparticle vaccines have been approved for clinical use⁴⁸, commercialization of this vaccine platform could be possible in the future. Moreover, the optimal components for a cocktail vaccine that are suggested by antigenic cartography may be platform-agnostic and could therefore be applied not just to other protein nanoparticle platforms, but also to other modalities such as mRNA-based vaccines. Indeed, the selection of an XBB strain (XBB.1.5) for incorporation in the recently approved monovalent vaccine is consistent with our results and it would be interesting to explore the protective efficacy of trivalent mRNA vaccines based on the compositions identified in this work against clade 1 sarbecoviruses.”

Reviewers' Comments:

Reviewer #3:

Remarks to the Author:

The changes to the manuscript are sufficient to answer my questions.